# MixReasoning: Switching Modes to Think

**Haiquan Lu** [1]   **Gongfan Fang** [1]   **Xinyin Ma** [1]   **Qi Li** [1]   **Xinchao Wang** [1]

## Abstract

Reasoning models enhance performance by tackling problems in a step-by-step manner, decomposing them into sub-problems and exploring long chains of thought before producing an answer. However, applying extended reasoning to every step introduces substantial redundancy, as sub-problems vary widely in difficulty and complexity: a small number of pivotal steps are genuinely challenging and decisive for the final answer, while many others only involve straightforward revisions or simple computations. Therefore, a natural idea is to endow reasoning models with the ability to adaptively respond to this variation, rather than treating all steps with the same level of elaboration. To this end, we propose MixReasoning, a framework that dynamically adjusts the depth of reasoning within a single response. MixReasoning enables fine-grained mode switching by training a lightweight concise LoRA adapter and controls its strength to trigger switches based on reasoning difficulty estimated from sliding-window token confidence, yielding human-like transitions between fast and slow reasoning. The resulting chain of thought then becomes a mixture of detailed reasoning on difficult steps and concise inference on simpler ones. Experiments on AIME24, MATH-500, GPQA, and GSM8K demonstrate that MixReasoning shortens reasoning length by 13%–49% across benchmarks of varying difficulty, delivering consistent efficiency gains while maintaining performance.

## 1. Introduction

Large Reasoning Models (LRMs) such as DeepSeek-R1 (Guo et al., 2025) and Qwen3 (Yang et al., 2025a) have exhibited notable effectiveness in solving complex tasks that involve multi-hop reasoning and logic-intensive inference. A key driver of these gains is the use of long chains of thought (CoTs) (Wei et al., 2022), which solve problems in a step-by-step manner by decomposing them into sub-problems and externalizing intermediate computations before arriving at a final answer. However, uniformly applying elaborate reasoning throughout the entire solution path often leads to overthinking (Chen et al., 2024; Pu et al., 2025), incurring substantial inference cost as thinking sequences become verbose and autoregressive decoding time scales roughly linearly with sequence length. The resulting latency and compute overhead are prohibitive for interactive applications and degrade user experience (Fu et al., 2024); moreover, excessively verbose traces hurt readability by introducing coherence filler and redundant self-checks that humans typically avoid (Fu et al., 2025a).

To mitigate these inefficiencies, a growing body of work has sought to improve reasoning efficiency by compressing the length of generated reasoning traces (Ma et al., 2025b; Chen et al., 2025; Ma et al., 2025a; Sun et al., 2025; Hao et al., 2024; Luo et al., 2025a). While effective at reducing latency, such global compression can inadvertently truncate pivotal reasoning steps, inducing underthinking (Su et al., 2025; Wang et al., 2025c) on critical subproblems and potentially degrading accuracy, making it difficult to maintain a favorable accuracy-efficiency balance. Another line of work adopts hybrid reasoning (Fang et al., 2025; Agarwal et al., 2025; Yang et al., 2025a), routing between reasoning and non-reasoning modes based on problem-level difficulty or user's tolerance. This improves the trade-off in some regimes, yet it assumes that a problem admits a clean binary partition (reason vs. no-reason) and leaves long-reasoning trajectories rife with redundancy: reasoning complexity is highly heterogeneous across substeps within one response, only a small set of pivotal steps, such as initial analysis, planning, decomposition, and key derivations, are genuinely difficult and thus demand more deliberate reasoning, whereas many other steps simply execute routine operations like arithmetic carry-outs, case enumeration, or straightforward transformations (Wang et al., 2025b).

By contrast, dual-process theories describe human cognition as involving two modes: a fast, intuitive mode (System 1) and a slow, deliberative mode (System 2). Crucially, people do not engage these modes uniformly; instead, they typically

---

[1]National University of Singapore, Singapore. Correspondence to: Xinchao Wang <xinchao@nus.edu.sg>.

*Proceedings of the $43^{rd}$ International Conference on Machine Learning*, Seoul, South Korea. PMLR 306, 2026. Copyright 2026 by the author(s).

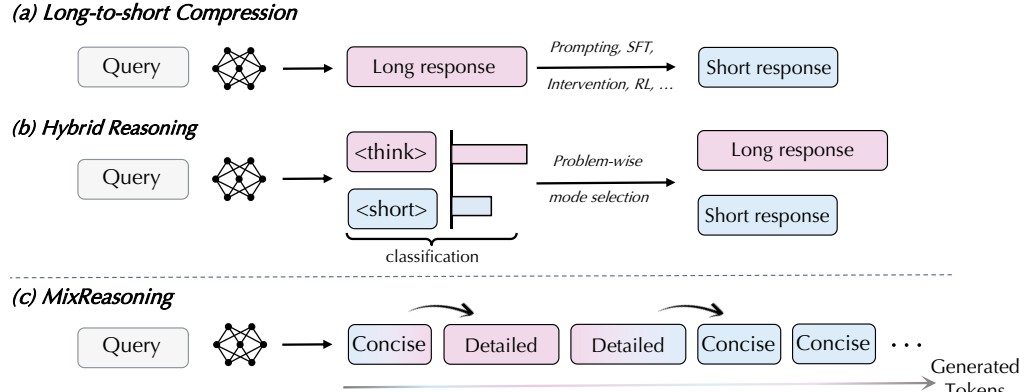

*Figure 1.* The comparison among Long-to-short compression, Hybrid reasoning, and MixReasoning. MixReasoning adaptively switches between different thinking modes during inference based on local uncertainty, achieving a balance of conciseness and detail.

think quickly when reasoning over easy and straightforward parts, and shift to slower, more effortful deliberation when encountering difficulty, via a conscious transition from System 1 to System 2 processing (Wason & Evans, 1974; Kahneman, 2011; Zhang et al., 2025b). This motivates us to explore whether we can endow reasoning models with the ability to adaptively **switch between fast and slow thinking modes within a single solution**, rather than applying a uniform level of deliberation across all steps. To achieve such reasoning ability, however, requires addressing two key challenges. First, the model must simultaneously support both detailed and concise reasoning, while allowing fine-grained, on-the-fly switching within one response, ideally without sacrificing the base model's deep reasoning capability. Second, the model must be able to identify which parts of the reasoning process are genuinely difficult and therefore warrant deeper, more elaborate thinking, as opposed to routine substeps that can be carried out succinctly.

Building on these insights, we propose MixReasoning, a framework that dynamically adjusts reasoning depth during generation to produce a mixture of detailed and concise reasoning. Specifically, MixReasoning enables intra-response mode switching by training a lightweight LoRA-based concise adapter and controlling its strength at inference time, thereby smoothly interpolating between detailed and concise reasoning without catastrophic forgetting. To determine when to switch, we leverage a difficulty-aware signal derived from the model's internal token distributions (Kang et al., 2025; Fu et al., 2025b). Intuitively, difficult reasoning steps tend to induce uncertainty in the next-token prediction, whereas routine execution often yields highly confident token choices. We operationalize this intuition with sliding-window token confidence, which aggregates token-level confidence over a short segment to obtain a more robust, semantically meaningful estimate of local step difficulty. Low-confidence windows consistently concentrate on analysis, planning, verification, and key derivations, while

high-confidence regions largely correspond to arithmetic carry-outs, case enumeration, and other straightforward follow-through. MixReasoning therefore triggers transitions to a more deliberative mode on low-confidence segments and reverts to concise mode when confidence recovers, allocating compute precisely where deeper reasoning is needed. Figure 1 illustrates the comparison among Long-to-short compression, Hybrid reasoning and MixReasoning.

**Contributions.** To summarize, our main contributions include: (1) We propose a simple yet effective mechanism for mode switching by training a lightweight LoRA-based concise adapter and controlling its strength at inference time, enabling a single served model to smoothly interpolate between detailed deliberation and concise inference without degrading the base reasoning capability. (2) We introduce sliding-window token confidence as a robust estimator of local reasoning difficulty, and show that the resulting difficulty distribution closely mirrors human problem-solving behavior. (3) We instantiate these components into MixReasoning, a framework that adaptively adjusts reasoning depth within a single response. Extensive experiments across models and benchmarks (AIME24, MATH-500, GPQA, GSM8K) demonstrate that MixReasoning can shorten reasoning length by 13%–49% and substantially improves inference efficiency while maintaining performance.

## 2. Related Work

**Efficient Reasoning.** LRMs adopt a structured problem-solving paradigm that decomposes complex problems into multi-step chains of thought (CoTs) (Wei et al., 2022), explicitly generating intermediate reasoning steps before the final answer. While this can boost accuracy, it also substantially increases inference-time compute due to long token sequences. To reduce this overhead, many methods aim to compress reasoning traces (Feng et al., 2025). Training-

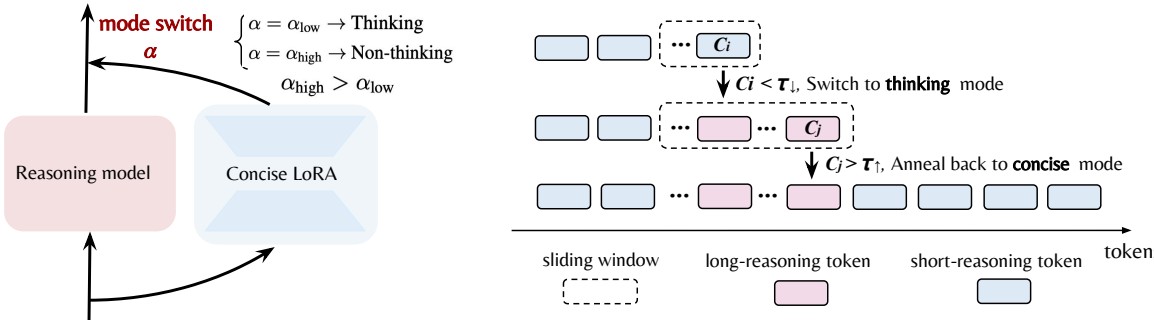

(a) LoRA strength-controlled mode switching

(b) Difficulty-aware switching via sliding-window token confidence

*Figure 2.* **MixReasoning overview.** A concise LoRA adapter is attached to a base reasoning model, and its strength $\alpha$ is scaled during inference to interpolate between detailed and concise reasoning; Sliding-window token confidence estimates local difficulty: low-confidence segments trigger a switch to thinking mode (with one-window rollback for re-generation), and confidence recovery switches back to concise mode via a hysteresis controller.

free approaches encourage models to "think less" (Renze & Guven, 2024; Ma et al., 2025a), intervene during decoding (Wang et al., 2025a;c; Tang et al., 2025), stop early once answer confidence is high (Yang et al., 2025b), or impose hard token budgets to bound rationale length (Sun et al., 2025). Training-based approaches include SFT on synthetic concise traces (Ma et al., 2025b) and RL with length-aware rewards that penalize long chains (Aggarwal & Welleck, 2025; Luo et al., 2025a). Although effective at lowering latency, these methods largely perform uniform compression across problems and steps, which can truncate pivotal reasoning and harm the accuracy–efficiency trade-off. Another direction is hybrid reasoning (Fang et al., 2025; Zhang et al., 2025b; Anthropic, 2025; Yang et al., 2025a), which routes by instance difficulty: easy queries receive short answers, while hard ones trigger long-form reasoning based on problem-level uncertainty or user's tolerance. This can reduce redundancy when many inputs admit straightforward solutions and preserve accuracy on truly difficult cases. However, hybrid routing does not address redundancy within long CoTs: models often remain verbose across routine substeps, and instance-level classification can be brittle since seemingly simple problems may contain localized hard segments. In contrast, our method aims to deliver consistent efficiency gains across varying problem difficulty while maintaining performance.

**Speculative Decoding and Reasoning.** Due to the memory-bound nature of LLM decoding, recent work has also leveraged the technique of speculation to accelerate model reasoning (Pan et al., 2025; Liao et al., 2025; Xia et al., 2024; Yang et al., 2025c; Li et al., 2025). Speculation interleaves a fast drafting step with verification by a larger target model, enforcing token-level or semantic-level agreement between a lightweight "draft" model and the base model. These methods reduce time per output token without necessarily shortening the CoT itself, and typically require co-serving both models, increasing memory footprint and operational complexity; consequently, rationales may remain verbose. This line of work is orthogonal to ours: MixReasoning shortens rationales via intra-CoT adaptive detail selection and can be combined with speculative decoding for additional speedups.

## 3. Method

In this section, we describe our methodology to enable LLMs to adaptively adjust reasoning depth within a single response, producing a mixture of detailed deliberation on difficult steps and concise inference on easy ones. MixReasoning is built around two key components:

1. **How to switch:** We enable switching between detailed and concise reasoning within a single served model by training a LoRA-based concise adapter and scaling its strength during inference.

2. **When to switch:** We trigger mode transitions based on step difficulty, estimated by sliding-window confidence derived from the model's internal token distributions.

### 3.1. Mode Switching via LoRA Strength Control

To enable intra-response adaptive reasoning, MixReasoning requires a control mechanism that can flexibly and reliably adjust reasoning depth without introducing a second model or degrading the base model's capability. We achieve this by training a lightweight LoRA adapter that distills a concise reasoning behavior into the base model while keeping the original "thinking mode" intact.

Formally, let $\theta$ denote the parameters of the base reasoning model. Given a question $q$ with a latent chain of thought $\mathbf{t}_{1:n}$ and final answer $a$, we construct short explanations $\mathbf{t}_{1:m}$ ($m < n$) that remain accurate and sufficient for producing

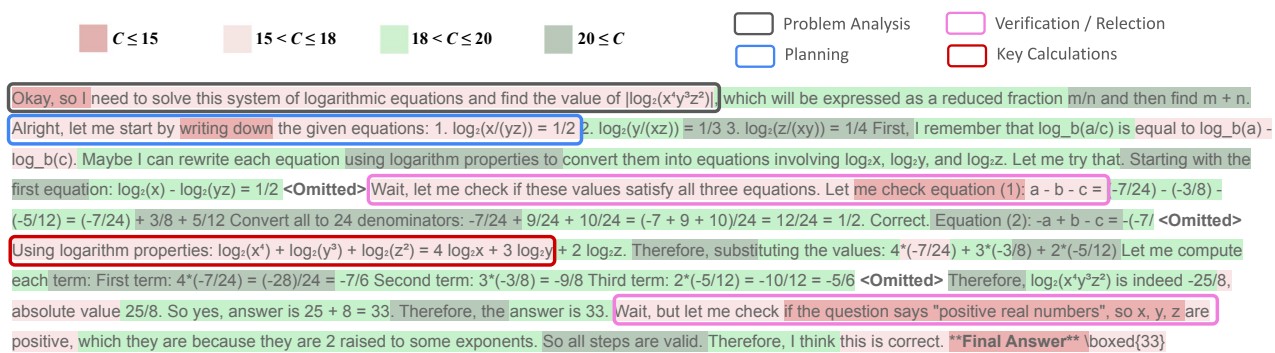

*Figure 3.* Sliding-window token confidence distribution in Qwen3-8B outputs. Low-confidence windows concentrate on analysis, planning, verification/reflection, and key calculations, whereas high-confidence regions largely correspond to routine execution, summarization, and straightforward logical follow-through, closely mirroring human problem-solving behavior.

$a$. We then fine-tune a LoRA update $\Delta\theta$ on such targets:

$$\max_{\Delta\theta} \ \mathbb{E}_{(q,a,\mathbf{t}_{1:m})\sim\mathcal{D}} \Big[ \log p_{\theta+\Delta\theta}(a \mid \mathbf{t}_{1:m}, q) \\ + \sum_{i=1}^{m} \log p_{\theta+\Delta\theta}(t_i \mid \mathbf{t}_{<i}, q) \Big]. \tag{1}$$

Unlike hybrid prompting or retraining-based approaches that entangle thinking and non-thinking behaviors in a single set of weights, our LoRA adapter preserves the base model and avoids catastrophic forgetting. Since multiple reasoning traces of varying length can lead to the same answer, $\Delta\theta$ can be interpreted as a task vector that primarily controls CoT length (Ilharco et al., 2022).

During inference (Fig. 2a), we **scale the LoRA strength** $\alpha$ to smoothly interpolate between detailed reasoning ($\alpha=0$) and concise mode ($\alpha>0$). Crucially, $\alpha$ can be adjusted at any decoding position, enabling fine-grained mode switching within a single response while serving a single model.

### 3.2. Difficulty-Aware Switching via Sliding-Window Token Confidence

Humans typically think quickly on easy, straightforward parts and engage slower, more deliberate reasoning when encountering difficulty, via a conscious shift from System 1 to System 2 processing (Kahneman, 2011; Zhang et al., 2025a). Motivated by this, and by recent findings that reasoning difficulty and quality can be predicted from metrics derived from a model's internal token distributions (Kang et al., 2025; Fu et al., 2025b), we use token confidence as the signal for when to switch modes. Concretely, at decoding step $t$, the model produces a distribution $p_t(\cdot)$ over the vocabulary. Let $\{v_{t,1}, \ldots, v_{t,k}\}$ denote the top-$k$ tokens under $p_t(\cdot)$. We define the token-level confidence score as the negative average log-probability of these top-$k$ candidates:

$$c_t \ = \ -\frac{1}{k} \sum_{j=1}^{k} \log p_t(v_{t,j}), \tag{2}$$

which can be interpreted as the negative log of the geometric mean over top-$k$ probabilities, i.e., $c_t = -\log\left(\prod_{j=1}^{k} p_t(v_{t,j})\right)^{1/k}$. Since the top-$k$ set typically captures most of the probability mass, a higher $c_t$ indicates a more peaked distribution (one candidate dominates while others are near zero), reflecting higher certainty. A more principled analysis of the token-level confidence metric can be seen in Appendix C.1.

However, single-token confidence is often sparse and unstable: a few extremely confident or uncertain tokens may mask longer low-confidence regions/segments, potentially hiding critical reasoning errors (Fu et al., 2025b; Pan et al., 2025). Moreover, the utility of intermediate substeps lies in the semantic insight they provide for downstream reasoning rather than the exact identity of individual tokens (Pan et al., 2025; Li et al., 2025), suggesting that difficulty should be assessed at the segment level. We therefore adopt a sliding-window token confidence:

$$C_t \ = \ \frac{1}{m} \sum_{j=t-m+1}^{t} c_j, \tag{3}$$

which yields a more semantically meaningful and robust signal. As illustrated in Fig. 3, low-confidence windows concentrate on analysis, planning, verification/reflection, and key calculations, precisely the steps humans deem difficult and decisive, whereas high-confidence regions typically correspond to routine execution, summarization, and straightforward logical follow-through. This pattern closely mirrors human problem-solving behavior.

Finally, we update the LoRA strength $\alpha$ online with a hys-

*Table 1.* Comparison of MixReasoning with fixed-mode baselines on four reasoning benchmarks. Concise mode uses a LoRA scaling parameter $\alpha$ to control compression strength, where larger $\alpha$ encourages shorter generations (stronger compression strength). We report Pass@1 and average generated tokens; Ori. denotes the base model without additional prompting, training, or switching.

| Models | AIME 2024 | | GPQA-Diamond | | Math-500 | | GSM8K | | Avg. | |
|---|---|---|---|---|---|---|---|---|---|---|
| | Pass@1 | #Token | Pass@1 | #Tokens | Pass@1 | #Tokens | Pass@1 | #Tokens | Pass@1 | #Tokens |
| **DeepSeek-R1-7B (Ori.)** | 52.00 | 10893 | 35.05 | 6030 | **90.24** | 3351 | 92.03 | 1214 | 67.33 | 5372 |
| **Concise Mode** $\alpha = 32$ | 48.00 | 8292 | 25.76 | 4536 | 84.32 | 2230 | 91.02 | 484 | 62.28 | 3885 |
| **Concise Mode** $\alpha = 64$ | 24.00 | 7297 | 30.32 | 4897 | 80.64 | 1725 | 88.42 | 424 | 55.85 | 3586 |
| **MixReasoning** | **53.33** | 8436 | **35.66** | 4991 | 89.76 | 2187 | **92.30** | 765 | **67.76** | 4094 (↓ 23 %) |
| **Qwen3-8B (Ori.)** | 64.67 | 11975 | **57.32** | 8624 | 92.30 | 5104 | 95.45 | 2403 | 77.44 | 7027 |
| **Concise Mode** $\alpha = 32$ | 52.13 | 9285 | 50.38 | 5918 | 85.34 | 2197 | 93.65 | 817 | 70.38 | 4554 |
| **Concise Mode** $\alpha = 64$ | 30.00 | 6021 | 45.56 | 5173 | 82.40 | 1535 | 92.89 | 534 | 62.71 | 3316 |
| **MixReasoning** | **67.33** | 8873 | 57.17 | 6734 | **92.76** | 3483 | **95.88** | 1389 | **78.29** | 5120 (↓ 28%) |
| **Qwen3-14B (Ori.)** | 69.34 | 12047 | **63.54** | 7161 | 93.84 | 4659 | 96.25 | 1950 | 80.72 | 6435 |
| **Concise Mode** $\alpha = 32$ | 53.33 | 7677 | 61.43 | 6502 | 89.64 | 1878 | 95.00 | 674 | 74.85 | 4217 |
| **Concise Mode** $\alpha = 64$ | 26.67 | 4534 | 51.01 | 2156 | 86.30 | 1071 | 94.87 | 372 | 64.71 | 2033 |
| **MixReasoning** | **71.33** | 9031 | 62.87 | 6225 | **94.34** | 3017 | **96.57** | 1137 | **81.28** | 4824 (↓ 25 %) |
| **QwQ-32B (Ori.)** | 65.53 | 11518 | 56.47 | 7923 | 93.80 | 4508 | 96.37 | 1745 | 78.04 | 6423 |
| **Concise Mode** $\alpha = 32$ | 53.33 | 8180 | 54.65 | 6078 | 92.90 | 2264 | 95.58 | 664 | 74.12 | 4296 |
| **Concise Mode** $\alpha = 64$ | 28.67 | 6902 | 48.89 | 3248 | 86.16 | 1238 | 95.14 | 391 | 64.72 | 2944 |
| **MixReasoning** | **67.33** | 8976 | **56.75** | 5783 | **94.30** | 3297 | **96.64** | 880 | **78.76** | 4734 (↓ 27 %) |

teresis controller (Fig. 2b):

$$\alpha_{t+1} = \begin{cases} \alpha_{\text{think}}, & C_t < \tau_{\text{low}} \ \wedge \ d_t \geq m, \\ \alpha_{\text{concise}}, & C_t > \tau_{\text{high}} \ \wedge \ d_t \geq m \end{cases} \quad (4)$$

where $d_t$ is the number of tokens generated since entering a new mode, and $m$ is the sliding-window size. This dwell constraint prevents rapid oscillations by enforcing at least one window of decoding before switching modes. In addition, when switching from concise to thinking mode, we rollback one window of tokens to re-generate the start of the low-confidence segment:

$$t \leftarrow \max(1, t - m + 1). \quad (5)$$

This allows think mode to correct early missteps and avoids committing to erroneous reasoning trajectories. The general pipeline of our method is illustrated in Appendix A.

### 3.3. KV-Cache Reuse and Serving Efficiency.

MixReasoning switches modes by adjusting only the LoRA strength $\alpha$ at inference time, without reloading or coordinating additional models. Crucially, mode switches reuse the KV cache already built for the destination mode: we only prefill the tokens generated in the other mode, avoiding recomputation over the shared prefix. Consequently, switching overhead scales with the length of the switched span and remains a small fraction of end-to-end latency. In practice, prefill is highly efficient (parallelizable and largely memory-bound) so even long prefills often take roughly the

wall-clock time of generating only 1–2 autoregressive tokens (Pan et al., 2025). We provide a detailed switching-cost analysis in Appendix C.3, and report end-to-end inference speedups over the base model in Sec. 4.5.

We also implement MixReasoning in vLLM (Kwon et al., 2023) to support efficient batched serving with LoRA strength control to switch modes. We expect this design to generalize to latency-sensitive reasoning workloads and to motivate further research in efficient adaptive inference.

## 4. Experiments

### 4.1. Experimental Setups

**Models and Benchmarks.** We conduct experiments on four widely used LRMs: DeepSeek-R1-7B (Guo et al., 2025), Qwen3-8B, Qwen3-14B (Alibaba, 2025), and QwQ-32B (Team, 2025). These models are known for their strong mathematical reasoning capabilities and long-chain-of-thought performance, and cover multiple parameter scales to test robustness. We evaluate them on AIME24 (Veeraboina, 2023), MATH-500 (Lightman et al., 2023), GSM8K (Cobbe et al., 2021), and GPQA (Rein et al., 2024). The first three benchmarks consist of mathematical tasks with varying levels of difficulty, while GPQA focuses on graduate-level STEM reasoning. All selected benchmarks are widely adopted in recent evaluations of state-of-the-art reasoning LLMs (Alibaba, 2025; Agarwal et al., 2025). We also include additional code-generation and commonsense reasoning benchmarks in Appendix C.4.

*Table 2.* Comparing MixReasoning with other advanced Long-to-Short CoT compression and hybrid reasoning methods.

| MODELS | TYPE | AIME 2024 | | GPQA-DIAMOND | | MATH-500 | | GSM8K | |
|---|---|---|---|---|---|---|---|---|---|
| | | PASS@1 | #TOKENS | PASS@1 | #TOKENS | PASS@1 | #TOKENS | PASS@1 | #TOKENS |
| **QWEN3-8B** | BASE LLM | 64.67 | 11975 | 57.32 | 8624 | 92.30 | 5104 | 95.45 | 2403 |
| **BECONCISE** (RENZE & GUVEN, 2024) | | 66.67 | 11371 | 57.23 | 7466 | 91.55 | 4232 | 95.68 | 1822 |
| **COT-VALVE** (MA ET AL., 2025B) | | 33.33 | 7412 | 40.76 | 5317 | 85.45 | 2375 | 88.46 | 784 |
| **NOWAIT** (WANG ET AL., 2025A) | SHORT COT | 64.83 | 9936 | 56.67 | 6575 | 92.25 | 3976 | 95.38 | 1406 |
| **DEER** (YANG ET AL., 2025B) | | 66.33 | 10298 | 55.45 | 7778 | 91.34 | 3259 | 95.62 | 1223 |
| **O1-PRUNER** (LUO ET AL., 2025A) | | 65.67 | 10498 | 56.95 | 8543 | 92.43 | 4968 | 95.56 | 1323 |
| **THINKLESS** (FANG ET AL., 2025) | HYBRID | 27.33 | 7099 | 24.75 | 8013 | 81.84 | 2555 | 84.18 | 624 |
| **MIXREASONING (OURS)** | ADAPTIVE | 67.33 | 8873 | 57.17 | 6734 | 92.76 | 3483 | 95.88 | 1389 |

**Training and Evaluation Details.** Our training procedure focuses on learning a concise-mode adapter via Low-Rank Adaptation (LoRA) (Hu et al., 2022). Specifically, we perform supervised fine-tuning (SFT) on the DeepScaleR-40K (Luo et al., 2025b) dataset, using short and correct solutions generated by non-thinking models (Yang et al., 2025a) as supervision. We adopt a standard LoRA configuration with rank $r = 32$ and scaling factor $\alpha_{\text{LoRA}} = 64$. This approach is highly effective: the resulting concise adapter consistently shortens the model's reasoning traces while mitigating catastrophic forgetting. For evaluation, we use vLLM inference framework and set the maximum generation length to 16,384 tokens. To reduce randomness, we perform five independent runs per query with a temperature of 0.6 and a top-$p$ value of 0.95, following the official recommendation (Yang et al., 2025a). We then compute the mean `pass@1` and the mean number of generated tokens across the five runs. More implementation details and hyperparameters settings can be found in Appendix B.

### 4.2. Main Results

Table 1 compares our method with the original reasoning model and its concise-mode variants, where $\alpha$ controls the LoRA adapter strength and larger values correspond to more aggressive compression. As shown, the reasoning model typically generates 2–5× more tokens than the concise model. Despite operating under a much smaller token budget, the concise model retains most of its performance on relatively easy benchmarks such as GSM8K and MATH-500, suggesting that the reasoning model often exhibits substantial redundancy and overthinking, and thus highlights the potential for significantly improving efficiency. However, on more challenging tasks such as AIME24, the concise model performs substantially worse and fails on a large fraction of problems (e.g., 30.00 vs. 64.67 on Qwen3-8B). This indicates that long-to-short compression methods introduce a harsh accuracy–latency trade-off: they either constrain the model's reasoning capacity or enforce a lower-quality generation mode uniformly across all steps, which often

leads to underthinking on difficult queries.

To address these limitations, MixReasoning adopts an adaptive strategy by dynamically switching between concise and reasoning modes based on the difficulty of reasoning steps. In our experiments, we use the base model ($\alpha = 0$) as the reasoning mode and set $\alpha = 64$ for the concise mode. Across all model backbones and benchmarks, MixReasoning reduces token usage by approximately 13%–49% while maintaining and in some cases even improving performance compared to the base reasoning model.

To further probe the robustness and generalization of MixReasoning, we provide additional evaluations in the appendix. Specifically, Appendix C.5 reports performance under Pass@$k$, showing that MixReasoning maintains strong performance beyond Pass@1 evaluation. Appendix C.4 further extends the evaluation to code-generation and commonsense reasoning benchmarks, including HumanEval (Chen et al., 2021), LiveCodeBench (Jain et al., 2024), CommonsenseQA (Talmor et al., 2019), and StrategyQA (Geva et al., 2021).

### 4.3. More Baseline Comparisons

To demonstrate the effectiveness of our approach, Table 2 presents a comparison between MixReasoning and several representative methods for efficient reasoning. Specifically, (i) **BeConcise** (Renze & Guven, 2024) is a prompting-based approach that encourages the model to produce shorter answers; (ii) **CoT-Valve** (Ma et al., 2025b) is an SFT-based technique that performs LoRA fine-tuning on mathematical reasoning datasets with short solutions; (iii) **O1-Pruner** is an RL-based method that prunes redundant reasoning steps; (iv) **Deer** (Yang et al., 2025b) adopts an early-exit strategy by terminating generation once the model becomes sufficiently confident; (v) **NoWait** (Wang et al., 2025a) improves efficiency by discouraging transition phrases such as "wait" and "alternatively"; and (vi) **Thinkless** (Fang et al., 2025) is a hybrid reasoning framework that selects different thinking modes based on problem-level difficulty.

As shown in Table 2, existing long-to-short compression and hybrid reasoning methods can reduce token usage on relatively easy benchmarks, often with only minor performance degradation. However, their gains diminish on challenging tasks such as AIME24 and GPQA: some methods achieve limited token savings and may even suffer notable accuracy drops, while some methods fall back to the full reasoning mode and effectively degenerate into the base model, offering little efficiency improvement. In contrast, MixReasoning remains consistently effective, reducing token usage while preserving strong performance. For example, on AIME24, MixReasoning reduces generation by 25% tokens while maintaining a high accuracy of 67.33.

## 4.4. Ablation Studies

In this section, we ablate the key design choices of MixReasoning, including the switching thresholds, the sliding-window size, and the switching criterion. These studies examine how different components of the difficulty-aware switching mechanism affect the performance of our method.

**Ablation of Threshold Selection.** MixReasoning relies on the sliding-window token confidence (Eq. 3) to estimate step-wise difficulty and trigger mode transitions. Concretely, when the confidence score falls below a low threshold $\tau_{\text{low}}$, we switch to thinking mode; when it rises above a high threshold $\tau_{\text{high}}$, we switch back to concise mode, as shown in Section 3.2 and Figure 2. A key question is how to choose $\tau_{\text{low}}$ and $\tau_{\text{high}}$. As shown in Fig. 4, the sliding-window confidence distribution is highly consistent across different benchmarks and model variants, concentrating in a similar numeric range with small variance. This empirical stability suggests that the switching thresholds do not require heavy per-dataset/model tuning and can be transferred with minimal performance instability.

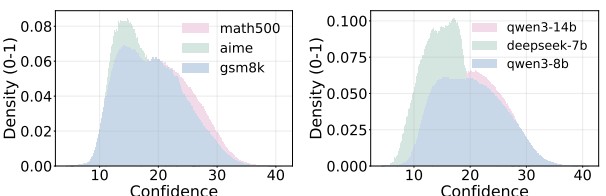

*Figure 4.* Sliding-window token confidence distributions across different benchmarks and models. The distributions largely overlap, suggesting stable and transferable switching thresholds.

To study its effect, we sweep $\tau_{\text{low}} \in \{12, 15, 18, 20\}$, corresponding roughly to the bottom $\{10\%, 20\%, 30\%, 40\%\}$ percentile regions of the confidence distribution, respectively. To reduce the tuning space, we set $\tau_{\text{high}} = \tau_{\text{low}} + 2$ throughout all experiments. Fig. 5 reports the resulting accuracy and generation length. As $\tau_{\text{low}}$ increases, more rea-

soning steps are classified as "difficult" and thus generated in thinking mode, leading to a higher thinking-mode ratio and longer responses. Importantly, this behavior provides a practical knob for controlling the performance-efficiency trade-off: smaller $\tau_{\text{low}}$ yields shorter responses suitable for low-latency instant QA, while larger $\tau_{\text{low}}$ allocates a larger token budget to thinking mode and improves robustness.

**Ablation of Sliding-Window Size.** The second important hyperparameter is the window size $m$ in Eq. 3. As shown in Figure 5, token-level confidence is often highly noisy and locally spiky: a few extremely confident or uncertain tokens may dominate the signal and lead to frequent, semantically ungrounded mode switches. Such unstable switching fragments coherent reasoning segments and can hurt both performance and response consistency. In contrast, sliding-window averaging smooths out local randomness and yields a more robust difficulty estimate, which better corresponds to meaningful reasoning phases such as planning, verification, and key calculations.

We conduct an ablation on window size and compare the sliding-window version against the single-token baseline ($m = 1$). As shown in Fig. 5, using single-token confidence leads to degraded performance due to unstable mode transitions, while moderate window sizes consistently improve accuracy and stability. Meanwhile, the exact choice of $m$ is not overly sensitive as long as it is not extremely small or excessively large (over-smoothing). In our main experiments, we set $m = 16$ as a balanced default.

**Ablation of Switching Criteria.** To verify that our switching criterion is the key driver of gains, we compare against alternative switching strategies. Specifically, we consider: (i) **Random switching**, which switches modes at random positions; (ii) **Reverse confidence**, which switches to thinking mode on high confidence (the opposite to our design); and (iii) **keywords triggering**, which switches to thinking mode when the model emits predefined thinking cue words (e.g., "Wait", "Hmm", and "Alternatively").

*Table 3.* Ablation of switching criteria on Qwen3-8B. We compare sliding-window confidence with Random, Reverse-Confidence, and Keywords triggering, reporting results on three datasets.

| DATASETS | AIME 2024 | | MATH-500 | | GSM8K | |
|---|---|---|---|---|---|---|
| | PASS@1 | #TOKENS | PASS@1 | #TOKENS | PASS@1 | #TOKENS |
| QWEN3-8B (ORI.) | 64.67 | 11975 | 92.30 | 5104 | 95.45 | 2403 |
| RANDOM | 43.67 | 8936 | 90.34 | 4073 | 94.32 | 1687 |
| REVERSE CONFIDENCE | 38.33 | 9343 | 87.40 | 3213 | 93.33 | 1974 |
| KEYWORDS | 60.17 | 9083 | 91.85 | 3849 | 95.16 | 2013 |
| MIXREASONING | 67.33 | 8873 | 92.76 | 3483 | 95.88 | 1389 |

Results in Table 3 show that our confidence-based switching consistently achieves the best performance. Random and reverse-confidence switching perform substantially worse,

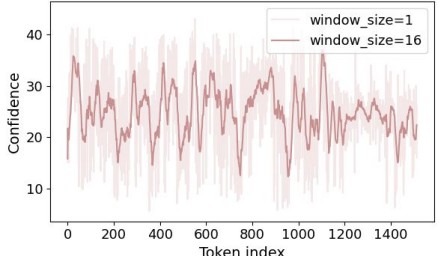 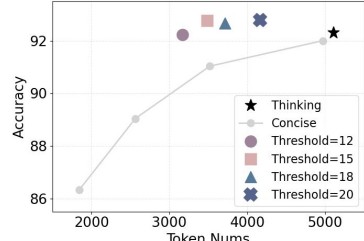 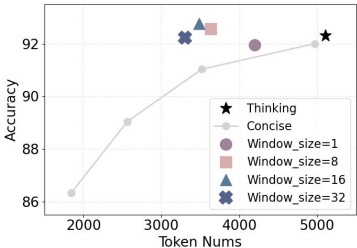

*Figure 5.* Ablation study on MixReasoning switching hyperparameters. Experiments are conducted on Qwen3-8B and evaluated on Math500 benchmark. *Left*: sliding-window token confidence distributions. *Middle*: performance and generation length under different low thresholds $\tau_{\text{low}}$. *Right*: sensitivity to the sliding-window size $m$.

demonstrating that gains are not simply from mixing two modes, but from placing thinking mode on the genuinely difficult parts. Thinking cue based switching captures partial uncertainty signals and can outperform random baselines, but remains less effective than sliding-window confidence, indicating that sliding-window token confidence provides a more robust difficulty indicator for adaptive reasoning. We also compare against alternative switching criteria derived from the model's internal token distributions (e.g., entropy, max probability, and margin) in Appendix C.2.

### 4.5. Further Analysis

In this section, we further analyze the behavior and practical efficiency of MixReasoning. We first examine how the model switches between concise and thinking modes across benchmarks, then analyze the concise-mode adapter, inference-time speedup, and qualitative examples.

**Switching Behavior Across Benchmarks.** We further analyze how MixReasoning switches between concise and thinking modes across benchmarks. As shown in Table 4, harder benchmarks require more thinking-mode tokens and more frequent mode switches. AIME24 and GPQA show higher thinking-mode ratios, 53.82% and 40.83%, with 66 and 38 switches, respectively. In contrast, MATH-500 and GSM8K use fewer thinking-mode tokens, 31.17% and 28.93%, with only 21 and 13 switches. This pattern is consistent with the difficulty of these benchmarks: AIME24 and GPQA contain more complex reasoning steps and decision-critical segments, which are more likely to trigger low-confidence regions and switch the model into thinking mode. By contrast, MATH-500 and GSM8K involve more routine computations and straightforward follow-through, allowing the model to remain in concise mode for a larger fraction of the response.

These results suggest that MixReasoning does not simply alternate between modes uniformly, but adaptively allocates detailed reasoning to more challenging datasets while maintaining concise generation on relatively simpler ones.

*Table 4.* Switching behavior of MixReasoning on Qwen3-8B across different benchmarks. We report the percentage of tokens generated in thinking mode and the total number of mode switches.

| Metrics | AIME24 | GPQA | MATH-500 | GSM8K |
|---|---|---|---|---|
| Thinking mode percentage (%) | 53.82 | 40.83 | 31.17 | 28.93 |
| Number of mode switches | 66 | 38 | 21 | 13 |

**Concise-Mode Adapter Fine-Tuning.** In our framework, we use LoRA fine-tuning to implement a concise-mode adapter, which is essential for enabling the model to generate both long-form reasoning traces and short responses. In this part, we provide additional implementation details and analyze how the choice of SFT data affects adapter quality. Specifically, we consider three datasets with increasing scale: (i) GSM8K (Cobbe et al., 2021), where we use the training split. It is a relatively easier mathematics-only dataset with very short solutions, often answer-only or with a minimal one–two-step rationale; (ii) DeepScaleR-40K (Luo et al., 2025b), a 40K math problem–solution pairs dataset sourced from AIME/AMC, Omni-MATH, and STILL; and (iii) OpenMathReasoning-300K (Moshkov et al., 2025), a 300K-instances mathematical reasoning dataset spanning a wide range of difficulty levels. For GSM8K, we directly use the ground-truth solutions. For DeepScaleR-40K and OpenMathReasoning-300K, we derive the corresponding short-form answers using Qwen3-14B-nonthinking (Yang et al., 2025a), a compact model optimized for concise responses. For fair comparison, we train the concise adapter on all datasets for a single epoch with the same LoRA configuration and optimization setup. Table 5 reports the fine-tuning results.

Interestingly, learning to produce short responses from a long-chain reasoning model is relatively easy: even with the smallest dataset (GSM8K), the target model quickly adapts to generate substantially shorter outputs. However, this compression often comes with a noticeable performance drop. Increasing the dataset scale partially narrows the performance gap, but the concise adapter still struggles on more challenging benchmarks (e.g., aime24). Moreover,

*Table 5.* Effectiveness of different SFT datasets for training a concise-mode adapter on Qwen3-8B. The three datasets span different scales: GSM8K (7K), DeepScaleR (40K), and OpenMath-Reasoning (300K) training examples.

| DATASETS | AIME 2024 | | MATH-500 | | GSM8K | |
|---|---|---|---|---|---|---|
| | PASS@1 | #TOKENS | PASS@1 | #TOKENS | PASS@1 | #TOKENS |
| QWEN3-8B (ORI.) | 64.67 | 11975 | 92.30 | 5104 | 95.45 | 2403 |
| GSM8K (TRAINING) | 26.66 | 7412 | 80.6 | 1751 | 82.52 | 217 |
| DEEPSCALER | 30.00 | 6021 | 82.40 | 1535 | 92.89 | 534 |
| OPENMATHREASONING | 27.99 | 5734 | 83.24 | 1662 | 93.50 | 537 |

scaling the distillation data from 40K to 300K yields only marginal improvements, suggesting that the limitations of long-to-short compression cannot be fully addressed by simply scaling the amount of training data.

**Inference-Time Speedup.** The primary overhead of MixReasoning comes from mode switching: when transitioning between modes, the model needs to prefill a small number of tokens generated in the other mode to maintain a consistent KV cache. Importantly, this cost is negligible in practice and scales linearly with the number of prefilling tokens required at each switch. As shown in Figure 6, the efficiency gains from reduced token generation are not offset by the switching overhead (Pan et al., 2025). Overall, MixReasoning consistently reduces end-to-end latency across all evaluated datasets relative to the base model. All end-to-end latency results are measured using vLLM and averaged over five runs. A detailed analysis of KV-cache reuse and switching overhead is provided in Appendix C.3.

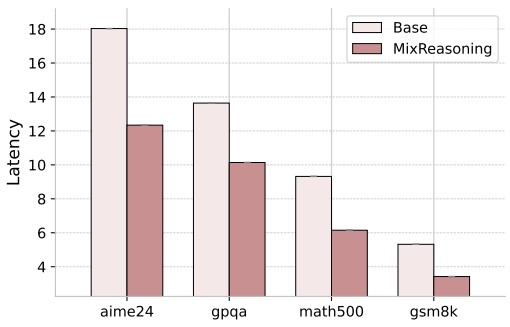

*Figure 6.* End-to-end latency of Qwen3-8B model on AIME24, Math500, GPQA-Diamond and GSM8K. The vLLM (Kwon et al., 2023) library is employed to perform inference.

**Qualitative Examples: Thinking Where It Matters.** We provide qualitative examples in Appendix D to illustrate how MixReasoning switches between concise and thinking modes within a single response. Compared with Long CoT, which often generates uniformly verbose traces with repeated self-checks and redundant reflections, MixReasoning concentrates detailed thinking on uncertain or decision-critical steps, such as problem interpretation, plan formation,

and key derivations. For more routine follow-through steps, it switches back to concise mode, avoiding unnecessary expansion and improving readability. This behavior is more consistent with natural reasoning patterns, where deliberation is concentrated around difficult parts rather than applied uniformly throughout the entire solution. The examples further show that the mixture between thinking and concise modes can be controlled by adjusting the confidence thresholds and window size, enabling a flexible trade-off between robustness and efficiency.

## 5. Conclusion

We present **MixReasoning**, a model-agnostic framework that enables LRMs to adaptively adjust reasoning depth within a single response. By switching between detailed and concise reasoning based on local difficulty, MixReasoning allocates computation to the truly difficult steps while compressing routine substeps. Experiments on AIME24, MATH-500, GPQA, and GSM8K show that MixReasoning consistently reduces reasoning tokens by 13%–49% across varying difficulty levels while maintaining task performance. Overall, our results suggest a simple principle for efficient reasoning: **think deeply only when it matters.**

## Acknowledgement

This project is supported by the National Research Foundation, Singapore, and Cyber Security Agency of Singapore under its National Cybersecurity R&D Programme and CyberSG R&D Cyber Research Programme Office (Award: CRPO-GC1-NTU-002).

## Impact Statement

MixReasoning improves the efficiency of large reasoning models by dynamically adjusting reasoning depth within a single response, reducing redundant tokens while maintaining performance, which can lower latency, compute cost, and energy use in interactive and resource-constrained settings. As with any more efficient inference technique, wider deployment may increase overall usage, so responsible application should follow standard practices for safe deployment (e.g., monitoring, filtering, and rate limiting) and encourage appropriate verification when concise reasoning is produced.

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

## A. Algorithm

---

**Algorithm 1** MixReasoning

---

**Input:** Base language model $p_\theta$; concise LoRA adapter $\Delta\theta$ (trained once via Eq. 1); maximum generation length $T_{\max}$; LoRA strengths $\alpha_{\text{think}}, \alpha_{\text{concise}}$; top-$k$ size $k$; window/dwell size $m$; thresholds $\tau_{\text{low}}, \tau_{\text{high}}$; prompt $q$.

**Output:** Output answer $y_{1:T}$ with adaptive reasoning depth.

1: Initialize mode $\leftarrow$ THINK; $\alpha \leftarrow \alpha_{\text{think}}$; $t \leftarrow 1$; $d \leftarrow 0$; buffer $\mathcal{B} \leftarrow \emptyset$
2: **while** not EOS **and** $t \leq T_{\max}$ **do**
3:      Use effective weights $\theta^\star \leftarrow \theta + \alpha\,\Delta\theta$          $\triangleright$ Single served model; scale adapter online
4:      Compute next-token distribution $p_t(\cdot) \leftarrow p_{\theta^\star}(\cdot \mid q, y_{<t})$
5:      Let $\{v_{t,1}, \ldots, v_{t,k}\}$ be top-$k$ tokens under $p_t(\cdot)$
6:      $c_t \leftarrow -\frac{1}{k}\sum_{j=1}^{k}\log p_t(v_{t,j})$          $\triangleright$ Token confidence, Eq. 2
7:      Push $c_t$ into $\mathcal{B}$ (keep last $m$); $C_t \leftarrow \frac{1}{|\mathcal{B}|}\sum_{c\in\mathcal{B}} c$          $\triangleright$ Sliding-window confidence, Eq. 3
8:      **if** $d \geq m$ **and** $C_t < \tau_{\text{low}}$ **and** mode = CONCISE **then**
9:          mode $\leftarrow$ THINK; $\alpha \leftarrow \alpha_{\text{think}}$; $d \leftarrow 0$          $\triangleright$ Hysteresis + dwell, Eq. 4
10:         $t \leftarrow \max(1, t-m+1)$; rollback last window of tokens in $y$      $\triangleright$ Re-generate low-confidence segment, Eq. 5
11:         **continue**
12:      **else if** $d \geq m$ **and** $C_t > \tau_{\text{high}}$ **and** mode = THINK **then**
13:         mode $\leftarrow$ CONCISE; $\alpha \leftarrow \alpha_{\text{concise}}$; $d \leftarrow 0$
14:      **end if**
15:      Sample $y_t \sim p_t(\cdot)$; append $y_t$; $t \leftarrow t+1$; $d \leftarrow d+1$
16:      Reuse KV-cache across switches by prefilling only the switched span          $\triangleright$ Sec. 3.3
17: **end while**
18: **return** $y_{1:T}$

---

## B. More Implementation Details

**Evaluation Metrics**    For all models and benchmarks, we use vLLM (Kwon et al., 2023) to generate responses on a single node with 4 H100 GPUs. We observe non-trivial variance in reasoning performance across random seeds (e.g., due to sampling and optimization stochasticity). To obtain stable and reliable comparisons, we therefore average over multiple runs and report the corresponding mean for pass@1 and the number of tokens. All results in Table 1 and Table 2 are reported as the mean over five independent runs. For all models and benchmarks, we set the maximum generation length to 16,384 tokens, with temperature $0.6$, top-$p$ $0.95$, and top-$k$ $20$. This configuration provides enough budget for long-form reasoning while keeping generation cost manageable.

For evaluation prompts, we adopt their default chat template for DeepSeek-R1-7B, Qwen3-8B, Qwen3-14B, and QwQ-32B. The prompt is adopted as follows.

> **Prompt for Evaluation (Math)**
>
> {Question}
> Please reason step by step, and put your final answer within \boxed{}.

The above prompt is used for math-oriented benchmarks, including GSM8K, AIME24, and MATH500. For multiple-choice QA benchmarks (e.g., GPQA-Diamond and CommonsenseQA), we use the following prompt format to ensure the model outputs a single explicit option:

> **Prompt for Evaluation (Multiple-choice QA)**
>
> {Question}
> {Options}
> Please reason step by step, and give your final answer in the format: The correct answer is <a single option letter from the provided choices>.

**Training Setting**   We use low-rank adaptation (LoRA) (Hu et al., 2022) to distill the non-thinking mode. Our default training set is derived from DeepScaleR (Luo et al., 2025b), a collection of ∼40K competition-style mathematics problems compiled from AIME (1984–2023), AMC, Omni-MATH, and Still. Each example contains a question, a worked solution, and a final answer enclosed in \boxed{}. We retain only the questions as prompts. For each query, we sample 10 short-chain responses from Qwen3-14B using the default sampling hyperparameters (Yang et al., 2025a). We then filter the candidates by answer correctness and select the shortest correct response as supervision; queries with no correct candidate are discarded.

All models are trained on a single node with four NVIDIA H100 GPUs. We set the global batch size to 64 and train for one epoch. The learning rate is $1 \times 10^{-5}$ with weight decay 0.01. For LoRA, we set the rank to 32 and the scaling factor $\alpha$ to 64. Training is implemented with Hugging Face `SFTTrainer` integrated with DeepSpeed ZeRO-2 optimization(Rasley et al., 2020).

## C. Further Theoretical and Empirical Analysis

In this section, we aim to provide a further theoretical and empirical analysis of the foundation and effectiveness of MixReasoning.

### C.1. Theoretical Justification of Difficulty-Aware Switching Signals

MixReasoning dynamically switches between thinking and concise modes using a local signal derived from the model's internal token distribution. While the controller is primarily motivated by empirical insights, we provide a lightweight theoretical justification of the switching signal. Specifically, we show that our token-level confidence admits a principled interpretation as a distributional concentration measure, rather than the conventional token-surprisal notion of confidence.

**Token-level confidence as peakiness of the top-$k$ distribution.**   As defined in Eq. (2),

$$c_t \;=\; -\frac{1}{k} \sum_{j=1}^{k} \log p_t(v_{t,j}),$$

which can be equivalently written as the negative log of the geometric mean over the top-$k$ probabilities, *i.e.*, $c_t = -\log \left( \prod_{j=1}^{k} p_t(v_{t,j}) \right)^{1/k}$. This form highlights that $c_t$ captures the shape of the local predictive distribution: when the distribution becomes sharply peaked (one candidate dominates while the remaining top-$k$ probabilities approach zero), the geometric mean decreases and $c_t$ increases. Since the top-$k$ set typically captures most of the probability mass, a larger $c_t$ indicates a more concentrated distribution and thus higher self-certainty, rather than merely reflecting the surprisal of a sampled token.

**Reverse-KL characterization and monotonicity under concentration.**   We further characterize $c_t$ via an exact reverse-KL decomposition and show that it increases whenever the top-$k$ distribution becomes more concentrated in the sense of majorization.

**Proposition C.1** (Token-level confidence as reverse-KL concentration). *Let $q_{t,j} = p_t(v_{t,j})$ be the top-k probabilities and $s_t = \sum_{j=1}^{k} q_{t,j}$ be the top-k probability mass. Define the normalized top-k distribution $\tilde{q}_{t,j} = q_{t,j}/s_t$ and the uniform distribution $U_k(j) = 1/k$. Then*

$$c_t \;=\; \log \frac{k}{s_t} \;+\; D_{\mathrm{KL}}(U_k \,\|\, \tilde{\mathbf{q}}_t)\,. \tag{6}$$

*Conditioned on $s_t$, $c_t(\tilde{\mathbf{q}}_t)$ is monotone under majorization: if $\tilde{\mathbf{q}}_t \succ \tilde{\mathbf{q}}'_t$, then*

$$c_t(\tilde{\mathbf{q}}_t) \geq c_t(\tilde{\mathbf{q}}'_t). \tag{7}$$

*Proof.* Using $q_{t,j} = s_t \tilde{q}_{t,j}$, we expand

$$c_t = -\frac{1}{k} \sum_{j=1}^{k} \log(s_t \tilde{q}_{t,j}) = -\log s_t - \frac{1}{k} \sum_{j=1}^{k} \log \tilde{q}_{t,j}.$$

By the definition of KL divergence,

$$D_{\mathrm{KL}}(U_k \| \tilde{\mathbf{q}}_t) = \sum_{j=1}^{k} \frac{1}{k} \log \frac{1/k}{\tilde{q}_{t,j}} = -\log k - \frac{1}{k} \sum_{j=1}^{k} \log \tilde{q}_{t,j}.$$

Substituting yields Eq. (6).

For the monotonicity claim, fix $s_t$ and define $f(x) = -\log x$, which is convex on $(0, 1]$. Then

$$c_t(\tilde{\mathbf{q}}_t) = \log \frac{k}{s_t} + \frac{1}{k} \sum_{j=1}^{k} f(\tilde{q}_{t,j}).$$

If $\tilde{\mathbf{q}}_t \succ \tilde{\mathbf{q}}'_t$, a standard characterization of majorization implies that there exists a doubly-stochastic matrix $A$ such that $\tilde{\mathbf{q}}'_t = A\tilde{\mathbf{q}}_t$. Hence for each $i$,

$$\tilde{q}'_{t,i} = \sum_{j=1}^{k} A_{ij}\tilde{q}_{t,j}, \qquad \sum_j A_{ij} = 1, \; A_{ij} \geq 0.$$

By Jensen's inequality,

$$f(\tilde{q}'_{t,i}) = f\Big( \sum_j A_{ij}\tilde{q}_{t,j} \Big) \leq \sum_j A_{ij} f(\tilde{q}_{t,j}).$$

Summing over $i$ and using $\sum_i A_{ij} = 1$ yields

$$\sum_{i=1}^{k} f(\tilde{q}'_{t,i}) \leq \sum_{j=1}^{k} f(\tilde{q}_{t,j}),$$

and therefore $c_t(\tilde{\mathbf{q}}_t) \geq c_t(\tilde{\mathbf{q}}'_t)$, which proves Eq. (7). $\qquad\square$

Eq. (6) shows that, up to the additive baseline $\log \frac{k}{s_t}$, $c_t$ equals a reverse-KL divergence from uniform. Because reverse-KL assigns equal weight to all top-$k$ candidates, it strongly penalizes near-zero probabilities, making $c_t$ sensitive to whether the distribution collapses to a single dominant token, consistent with the notion of self-certainty. Eq. (7) further guarantees that $c_t$ increases whenever the top-$k$ distribution becomes more concentrated, rather than merely reflecting token-level NLL.

### C.2. Relation to other Distribution-Derived Metrics

Having established $c_t$ as a principled concentration measure (reverse-KL from uniform), we now compare it with common distribution-derived alternatives. Beyond conceptual differences, we also empirically verify that our switching criterion is effective by comparing against alternative switching strategies based on these distribution-derived metrics (Table 6).

**Entropy (Shannon uncertainty).**    A common choice is the top-$k$ (normalized) entropy (Shannon, 1948; Lewis, 1995)

$$H(\tilde{\mathbf{q}}_t) = -\sum_{j=1}^{k} \tilde{q}_{t,j} \log \tilde{q}_{t,j} = \log k - D_{\mathrm{KL}}(\tilde{\mathbf{q}}_t \| U_k). \tag{8}$$

Entropy decreases as the distribution becomes more peaked, whereas our score increases with peakedness. This difference is not merely a sign flip: $D_{\mathrm{KL}}(U_k \| \tilde{\mathbf{q}}_t)$ penalizes small probabilities much more strongly than $D_{\mathrm{KL}}(\tilde{\mathbf{q}}_t \| U_k)$ because it weights each candidate equally. As a result, $c_t$ is particularly sensitive to "nearly-impossible" alternatives within the top-$k$, making it a sharper detector of local decisiveness.

**Max probability.** Another proxy is $\max_j q_{t,j}$ (least-confidence uncertainty) (Settles, 2009). While simple, it ignores the rest of the top-$k$ structure: two distributions can share the same max probability but differ substantially in how mass is spread among competitors. In contrast, $c_t$ depends on the entire top-$k$ set and thus captures ambiguity among multiple plausible next tokens.

**Margin.** A margin proxy uses either $q_{t,1} - q_{t,2}$ or $\log(q_{t,1}/q_{t,2})$ (Tong & Koller, 2001). Our score can be viewed as a multi-way generalization of margin: it aggregates the log-probabilities of multiple competitors, and is therefore sensitive not only to the top-1 vs top-2 gap but also to whether the distribution "collapses" beyond the first few tokens. For $k = 2$, letting $q_{t,1} = \frac{s_t}{2}(1 + \delta)$ and $q_{t,2} = \frac{s_t}{2}(1 - \delta)$ gives

$$c_t = \log \frac{2}{s_t} + \frac{1}{2} \log \frac{1}{1 - \delta^2}, \tag{9}$$

which increases monotonically with $|\delta|$ (i.e., larger margin implies higher $c_t$). For larger $k$, $c_t$ generalizes this behavior by considering multiple competitors simultaneously.

**Empirical Comparison as Switching Criteria.** To verify that our switching criterion is effective in practice, we instantiate alternative switching strategies by replacing our sliding-window confidence with entropy, max probability, and margin derived from the same top-$k$ token distribution. We report Pass@1 and generation length (#Tokens) on AIME 2024, Math-500, and GSM8K, as summarized in Table 6.

*Table 6.* Ablation of switching criteria on Qwen3-8B. To verify the effectiveness of our switching rule, we compare sliding-window confidence with alternative switching strategies derived from the model's internal token distributions (Entropy, Max probability, and Margin), reporting Pass@1 and generation length (#Tokens) on three datasets.

| DATASETS | AIME 2024 | | MATH-500 | | GSM8K | |
|---|---|---|---|---|---|---|
| | PASS@1 | #TOKENS | PASS@1 | #TOKENS | PASS@1 | #TOKENS |
| QWEN3-8B (ORI.) | 64.67 | 11975 | 92.30 | 5104 | 95.45 | 2403 |
| ENTROPY | 64.17 | 9784 | 91.85 | 4467 | 95.38 | 1498 |
| MAX PROBABILITY | 63.33 | 10384 | 92.15 | 3984 | 95.65 | 1787 |
| MARGIN | 63.83 | 9987 | 91.95 | 4109 | 95.93 | 1918 |
| MIXREASONING | 67.33 | 8873 | 92.76 | 3483 | 95.88 | 1389 |

**Connections to Confidence-Based Test-Time Selection** Finally, our analysis aligns with recent works that use internal confidence signals for test-time scaling and compression. **Self-Certainty** (Kang et al., 2025) aggregates distributional signals to perform reward-free best-of-$N$ selection, demonstrating that internal confidence correlates with correctness across reasoning tasks. **DeepConf** (Fu et al., 2025b) leverages local confidence measurements to filter or early-stop low-quality traces, yielding substantial token savings without additional training. These results support our premise that internal distribution-derived confidence can serve as an effective control signal for adaptive computation at inference time.

## C.3. Switching Overhead

MixReasoning switches modes by changing only the LoRA strength $\alpha$ online, i.e., $\theta^\star = \theta + \alpha\,\Delta\theta$. While the backbone is shared, the KV cache is mode-dependent because $\alpha_{\text{think}} \neq \alpha_{\text{concise}}$ yields different activations. Hence, after switching from mode $a$ to mode $b$, we can reuse the KV cache already built for mode $b$ on the shared prefix, and we only need to prefill the tokens generated in mode $a$ since the last time we visited mode $b$.

Let $\ell$ be the length of this switched span. The switching overhead is exactly the cost of one prefill:

$$\Delta T_{\text{switch}}(\ell) \;=\; \mathcal{P}(\ell), \tag{10}$$

where $\mathcal{P}(\ell)$ denotes the wall-clock latency of prefilling $\ell$ tokens under the destination mode. Over a full response of length $T$ with $S$ switches and switched spans $\{\ell_i\}_{i=1}^{S}$, total latency is

$$T_{\text{mix}} \;=\; \mathcal{D}(T) \;+\; \sum_{i=1}^{S} \mathcal{P}(\ell_i), \tag{11}$$

where $\mathcal{D}(T)$ is the autoregressive decoding cost for generating $T$ tokens under the adaptive schedule.

Our dwell/window size $m$ (Alg. 1) prevents frequent oscillations, so switches cannot occur more often than roughly once every $m$ tokens (except rare rollback events), yielding $S \lesssim \lceil T/m \rceil$. Defining the relative switching latency as $\sum_{i=1}^{S} \mathcal{P}(\ell_i)/\mathcal{D}(T)$, and noting that prefill is highly parallelized and efficient in practice, so can be upper bounded by a small constant number of decode steps (i.e., $\mathcal{P}(\ell) \leq \gamma \mathcal{D}(1)$ with $\gamma \approx 1$–$2$) (Pan et al., 2025), we obtain an upper bound of approximately $\gamma/m$. With a moderate $m$, the relative latency contributed by mode switching is therefore small, making the switching overhead negligible in practice.

### C.4. Additional Reasoning Benchmarks

To further examine downstream task diversity, we complement the primary experiments with additional evaluations on code-generation and commonsense reasoning benchmarks. Specifically, we evaluate MixReasoning on HumanEval (Chen et al., 2021) and LiveCodeBench (Jain et al., 2024) for code generation, and on StrategyQA (Geva et al., 2021) and CommonsenseQA (Talmor et al., 2019) for commonsense reasoning. These tasks broaden the evaluation beyond the primarily math- and science-focused benchmarks used in the main paper.

CommonsenseQA is a natural-language multiple-choice QA benchmark with five answer options, designed to require commonsense knowledge and multi-hop associative reasoning rather than mathematical derivations. StrategyQA further tests implicit multi-step commonsense reasoning through yes/no questions. HumanEval and LiveCodeBench evaluate functional code generation, where models must produce executable programs that pass hidden or public test cases. Together, these benchmarks provide a more diverse evaluation of whether MixReasoning generalizes beyond STEM-style reasoning tasks.

We use the same evaluation settings as in the main paper, including identical decoding hyperparameters across all compared methods. As shown in Table 7, MixReasoning consistently reduces generation length compared with the original reasoning model while preserving or improving task performance across both code-generation and commonsense reasoning benchmarks. In particular, MixReasoning improves HumanEval Pass@1 from 91.95 to 93.29 while reducing the average generation length from 2881 to 2643 tokens. On CommonsenseQA, it improves accuracy from 83.39 to 83.90 while reducing the average generation length from 1183 to 895 tokens. These results provide additional evidence that MixReasoning delivers efficiency gains without sacrificing reasoning quality, and that its benefits extend beyond mathematical domains.

*Table 7.* Additional code-generation and commonsense reasoning benchmark results. Results are evaluated on Qwen3-8B. We report Pass@1 and the average number of generated tokens for all benchmarks.

| Models | HumanEval | | LiveCodeBench | | StrategyQA | | CommonsenseQA | |
|---|---|---|---|---|---|---|---|---|
| | Pass@1 | #Tokens | Pass@1 | #Tokens | Pass@1 | #Tokens | Pass@1 | #Tokens |
| **Qwen3-8B (Ori.)** | 91.95 | 2881 | 44.34 | 8352 | 77.32 | 1304 | 83.39 | 1183 |
| **Concise Mode** | 90.93 | 2543 | 46.18 | 7289 | 74.97 | 989 | 74.87 | 703 |
| **MixReasoning** | 93.29 | 2643 | 45.00 | 7689 | 77.04 | 1184 | 83.90 | 895 |

### C.5. Performance Evaluation under Pass@k

While our main experiments report Pass@1 performance, Pass@k provides a complementary perspective on test-time scaling by measuring whether at least one of $k$ sampled reasoning trajectories produces the correct answer. This metric is particularly useful for evaluating whether a method can improve not only the most likely response, but also the diversity and quality of candidate reasoning paths under repeated sampling. Therefore, we further report Pass@2–5 on AIME24 and MATH-500 using Qwen3-8B.

As shown in Table 8, MixReasoning consistently improves Pass@k performance on the more challenging AIME24 benchmark, with gains increasing from Pass@2 to Pass@5. On MATH-500, where the original model already achieves strong performance, MixReasoning still yields improvements under Pass@2–4 and remains comparable at Pass@5. These results indicate that MixReasoning can better exploit test-time sampling by producing more effective reasoning trajectories, further demonstrating the robustness of our method beyond the standard Pass@1 setting.

*Table 8.* Pass@2–5 performance of Qwen3-8B and MixReasoning on AIME24 and MATH-500.

| Models | AIME24 | | | | Math500 | | | |
|---|---|---|---|---|---|---|---|---|
| | Pass@2 | Pass@3 | Pass@4 | Pass@5 | Pass@2 | Pass@3 | Pass@4 | Pass@5 |
| **Qwen3-8B (Ori.)** | 71.33 | 72.00 | 72.67 | 73.33 | 93.60 | 94.50 | 94.65 | 95.12 |
| **MixReasoning** | 73.67 | 76.67 | 78.67 | 80.00 | 94.12 | 94.66 | 94.88 | 95.00 |

## D. Qualitative Examples: Thinking Where It Matters

We provide qualitative examples to illustrate the behavior of MixReasoning on representative reasoning problems, including a more challenging case that requires careful intermediate deliberation. As shown in Figures 7 and 8, MixReasoning produces reasoning traces that are substantially more focused than Long CoT while still preserving answer correctness. Compared with Long CoT, which often generates uniformly verbose reasoning with repeated self-checks, redundant restatements, and excessive reflection, MixReasoning allocates detailed thinking only to steps where additional deliberation is needed. For relatively straightforward parts of the solution, it switches back to concise mode, avoiding unnecessary expansion. This behavior leads to shorter and more readable traces, while retaining sufficient reasoning depth around ambiguous or decision-critical steps. The second example Figure 8 further highlights this advantage on a harder problem. In this case, Long CoT exhibits extended self-reflection and verbose intermediate explanations. Such traces can be less consistent with natural human reasoning patterns, where effort is typically concentrated around the most difficult or uncertain steps rather than distributed uniformly throughout the entire solution. By contrast, MixReasoning reduces excessive verbosity and redundant self-reflection, focusing its reasoning budget on the core challenging steps.

Importantly, MixReasoning also provides explicit control over the mixture ratio between thinking and concise modes as shown in Figure 7. By adjusting the confidence thresholds $(\tau_{\text{low}}, \tau_{\text{high}})$ and the window size $m$, we can trade off efficiency and robustness: more aggressive settings allocate a larger fraction of tokens to thinking mode and produce more detailed rationales, whereas more conservative settings keep responses compact and readable.

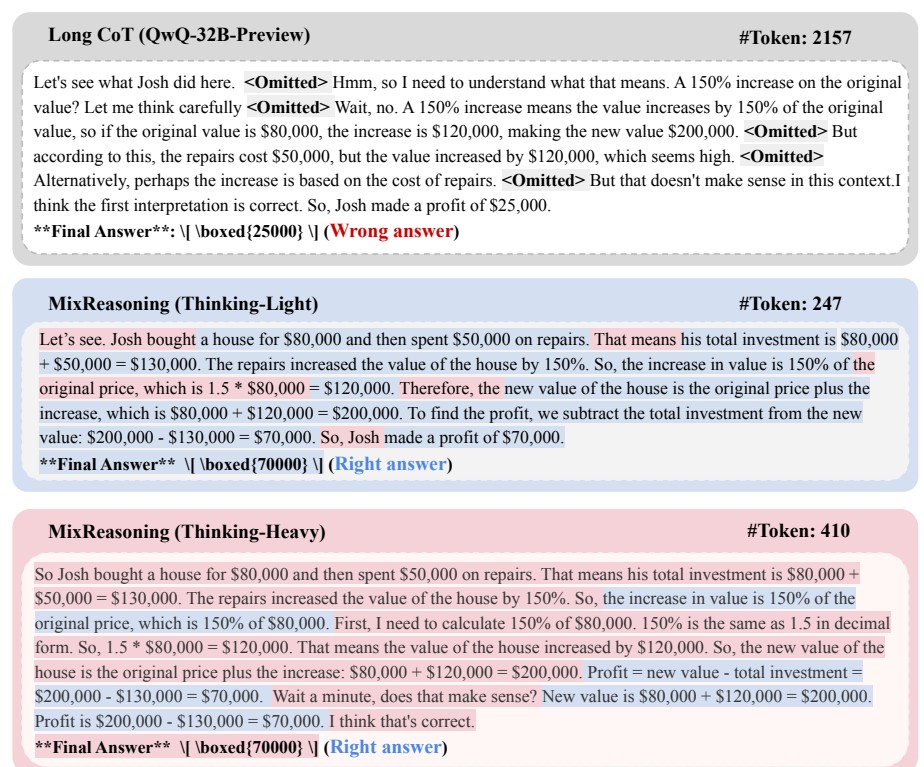

*Figure 7.* Qualitative comparison of reasoning traces. Long CoT is verbose, while MixReasoning focuses detailed thinking on high-uncertainty steps and stays concise elsewhere. Thinking-mode tokens are highlighted in red and concise-mode tokens in blue .

***Question:*** Bill walks $\frac{1}{2}$ mile south, then $\frac{3}{4}$ mile east, and finally $\frac{1}{2}$ mile south. How many miles is he, in a direct line, from his starting point? Express your answer as a decimal to the nearest hundredth."

---

**Long CoT (Qwen3-8B)**          **#Token: 2216**

Okay, so Bill is walking in different directions, and I need to figure out how far he is from his starting point. Let me try to visualize this.

First, he walks 1/2 mile south. Then he goes 3/4 mile east, and finally another 1/2 mile south. Hmm. So maybe I should draw a coordinate system to track his movements? Let me think. If I consider his starting point as the origin (0,0), then moving south would be in the negative y-direction, and east would be the positive x-direction.

Let me write down each step:

1. First, he walks 1/2 mile south. So starting at (0,0), moving south (negative y-direction) by 1/2 mile. That would put him at (0, -1/2).

2. Then he walks 3/4 mile east. East is the positive x-direction, so adding 3/4 to his x-coordinate. So his new position is (3/4, -1/2).

3. Finally, he walks another 1/2 mile south. So again, moving in the negative y-direction. Adding another -1/2 to his y-coordinate. So his final position is (3/4, -1/2 - 1/2) = (3/4, -1).

Now, I need to find the straight-line distance from his starting point (0,0) to his final position (3/4, -1).

The distance formula between two points (x1, y1) and (x2, y2) is sqrt[(x2 - x1)^2 + (y2 - y1)^2].

Plugging in the values:

Distance = sqrt[(3/4 - 0)^2 + (-1 - 0)^2] = sqrt[(9/16) + 1].

Wait, let me compute that.

First, (3/4)^2 is 9/16. Then (-1)^2 is 1. So adding those together: 9/16 + 1. To add them, I need a common denominator. 1 is 16/16, so total is 9/16 + 16/16 = 25/16.

Therefore, the distance is sqrt(25/16). The square root of 25/16 is 5/4.

But 5/4 is 1.25. Wait, but the question says to express the answer as a decimal to the nearest hundredth. 1.25 is already to the nearest hundredth, so that would be 1.25.

Wait, but let me double-check my steps. Maybe I made a mistake in coordinates?

Starting at (0,0). South is negative y. So first move: (0, -1/2). Then east is positive x: (3/4, -1/2). Then another south: (3/4, -1/2 - 1/2) = (3/4, -1). That seems correct.

Then distance from (0,0) is sqrt( (3/4)^2 + (-1)^2 ). Which is sqrt(9/16 + 1). Which is sqrt(25/16) = 5/4 = 1.25.

Hmm. So 1.25 is already exact. So as a decimal, it's 1.25. But maybe I need to check if there's a miscalculation?

Alternatively, maybe I should convert the fractions to decimals first?

Let me try that approach.

First movement: 1/2 mile south is 0.5 miles. Then 3/4 mile east is 0.75 miles. Then another 0.5 miles south.

So, in terms of coordinates:

Starting at (0,0). After first move, (0, -0.5). Then east 0.75: (0.75, -0.5). Then south 0.5: (0.75, -0.5 - 0.5) = (0.75, -1.0).

Distance from (0,0) is sqrt( (0.75)^2 + (-1.0)^2 ) = sqrt(0.5625 + 1) = sqrt(1.5625).

What is sqrt(1.5625)? Well, 1.25 squared is 1.5625 because 1.25 * 1.25 = 1.5625. So that's 1.25.

So same answer. Therefore, 1.25 miles.

But wait, the problem says "Express your answer as a decimal to the nearest hundredth." Since 1.25 is already to the nearest hundredth, there's no need to round. So the answer is 1.25.

But let me check again. Maybe I made a mistake in the direction? For example, is moving south and then east and then south... Maybe I should think of it as a right triangle?

So, if he goes south 1/2, then east 3/4, then south 1/2. So total south movement is 1/2 + 1/2 = 1 mile, and east movement is 3/4 mile. So the displacement is a right triangle with legs 3/4 and 1. Therefore, the distance is sqrt( (3/4)^2 + 1^2 ) = same as before.

Yes, so that's sqrt(9/16 + 16/16) = sqrt(25/16) = 5/4 = 1.25.

Therefore, I think the answer is 1.25.

But let me check if there is any alternative interpretation. For example, maybe the problem is in three dimensions? But no, it's all in two dimensions, south and east. So no, that's not the case.

Alternatively, **<Omitted>** Alternatively, **<Omitted>** Alternatively, **<Omitted>** But maybe the problem is **<Omitted>** But just to be thorough, **<Omitted>** But just to make sure, let me check with another approach. **<Omitted>** But just to check if there's any possible mistake in interpreting directions. <Omitted> Alternatively, <Omitted> Alternatively, <Omitted> Alternatively, **<Omitted>** But just to make sure, **<Omitted>** But no, **<Omitted>** Alternatively, **<Omitted>** But no, **<Omitted>**

Therefore, I think my answer is correct.

**Final Answer** \boxed{1.25} **(Right answer)**

---

**MixReasoning**          **#Token: 941**

Okay, so Bill is walking in different directions, and I need to figure out how far he is from his starting point in a straight line. Let me try to visualize this.

First, he walks 1/2 mile south. Then he turns east and walks 3/4 mile. Then he walks another 1/2 mile south. Hmm, okay. Let me think about how to represent this on a coordinate system.

Maybe I can use coordinates to track his position.

Let me set up a coordinate system where the starting point is at (0, 0). Let's assume that north is the positive y-direction and east is the positive x-direction. So, moving south would be in the negative y-direction, and east is positive x.

Starting at (0, 0). First, he walks 1/2 mile south. That would take him to (0, -1/2). Then he walks 3/4 mile east, so that's adding 3/4 to the x-coordinate. So his new position is (3/4, -1/2). Then he walks another 1/2 mile south. That would subtract another 1/2 from the y-coordinate. So his final position is (3/4, -1/2 - 1/2) = (3/4, -1).

Now, I need to find the straight-line distance from his starting point (0, 0) to his final position (3/4, -1).

The distance formula between two points (x1, y1) and (x2, y2) is sqrt[(x2 - x1)^2 + (y2 - y1)^2].

Plugging in the values, that would be sqrt[(3/4 - 0)^2 + (-1 - 0)^2] = sqrt[(9/16) + 1].

Wait, sqrt[(9/16) + 1]. Let me compute that. 1 is 16/16, so adding 9/16 gives 25/16. Then sqrt(25/16) is 5/4.

Wait, 5/4 is 1.25. So the straight-line distance is 1.25 miles? Let me check my steps again to make sure I didn't make a mistake.

Starting at (0, 0). First move: 1/2 mile south to (0, -1/2). Then 3/4 mile east to (3/4, -1/2). Then 1/2 mile south again: subtract 1/2 from the y-coordinate. So -1/2 - 1/2 is -1. So yes, (3/4, -1).

Then distance from (0, 0) is sqrt( (3/4)^2 + (-1)^2 ).

Calculating (3/4)^2 is 9/16. (-1)^2 is 1. So total is 9/16 + 16/16 = 25/16. Square root of 25/16 is 5/4, which is 1.25. So 1.25 miles.

But the problem says to express the answer as a decimal to the nearest hundredth. 5/4 is exactly 1.25, so that's already to the nearest hundredth. So the answer is 1.25.

Wait, but let me just make sure that I didn't mix up any directions. When he walks south, it's negative y, then east is positive x. Then another south, so more negative y. So the coordinates are correct.

Alternatively, maybe I can think of it as a right triangle. The eastward movement is 3/4 mile, and the total southward movement is 1/2 + 1/2 = 1 mile. So the legs of the right triangle are 3/4 and 1. Then the hypotenuse is sqrt( (3/4)^2 + 1^2 ) which is sqrt(9/16 + 1) = sqrt(25/16) = 5/4. Yep, same result. So that's 1.25.

Therefore, the answer is 1.25 miles.

**Final Answer** \boxed{1.25} **(Right answer)**

---

*Figure 8.* Qualitative comparison of reasoning traces. Thinking-mode tokens are highlighted in red and concise-mode tokens in blue.

