# OpenReview forum: "MixReasoning: Switching Modes to Think"
_ICML.cc/2026/Conference — ICML 2026 regular_

### Official Review · Reviewer_YdxE · 2026-02-25

**Soundness:** 2
**Presentation:** 3
**Significance:** 2
**Originality:** 3
**Overall Recommendation:** 3
**Confidence:** 3

**Summary:**

Large language models (LLMs) often generate relatively long responses during the reasoning process, which contain many complex reasoning steps and detailed thought processes. This lengthy reasoning process significantly reduces the model’s reasoning efficiency and may even lead to performance degradation due to overthinking and related issues.

To address this issue, the authors proposed the MixReasoning, which is a framework to allow the model to determine its own reasoning intensity and decide whether to engage in deep thinking. Concretely, The authors use LoRA modules to train the model, enabling it to acquire hybrid reasoning capabilities, that is, allowing it to autonomously decide the intensity of reasoning to apply. To determine when to perform deep reasoning, the authors rely on attention scores within a sliding window as a criterion during the training process.

To verify the effectiveness of the proposed method, the authors conduct evaluations on multiple tasks. The experimental results demonstrate that MixReasoning can reduce the reasoning length while maintaining the model’s reasoning performance.

**Compliance With Llm Reviewing Policy:**

Affirmed.

**Final Justification:**

In the authors’ response, the model performance exhibits substantial variance, making it difficult to demonstrate the effectiveness of the proposed method. I maintain a weak reject assessment.

**Key Questions For Authors:**

Please explain and address the concerns in weaknesses part.

**Limitations:**

+ The authors need to include more downstream evaluation tasks and conduct experiments on a broader range of models (with different scales and from different families) to strengthen the persuasiveness of the experimental results.
+ Further elaborate on the motivation and theoretical foundation of the method.
+ Discuss the potential real-world application scenarios of the method, whether there is room for further optimization, and whether it can assist the training process.

**Strengths And Weaknesses:**

**Strengths**
+ The authors propose MixReasoning, which achieves performance comparable to slow-thinking systems while conducting the reasoning process more efficiently.
+ The authors control the depth of reasoning at a finer granularity, enabling the model to better regulate its reasoning process.
+ The authors employ LoRA modules for training, reducing computational overhead and further improving the efficiency of the proposed algorithm.
+ The authors evaluate the method on four downstream tasks and compare it with multiple baseline methods, providing preliminary evidence of its effectiveness.

**Weaknesses**
+ The method of determining whether to engage in deep thinking based on Sliding-Window Token Confidence lacks a solid theoretical foundation, and its motivation is not sufficiently convincing.
+ The number of evaluation tasks and base models is relatively limited, rendering the experimental validation insufficient and weakening the overall persuasiveness of the results.
+ Can this method be applied to end-to-end RL training to further enhance the model’s reasoning ability and efficiency? The potential application value of this method is not discussed in the paper.

---

> ### Author Rebuttal · Authors · 2026-03-31
>
> We thank the reviewer for the thoughtful feedback and recognizing the efficiency, fine-grained reasoning control, and empirical effectiveness of MixReasoning. We respond to critiques and comments below.
>
> ---
> >Weakness 1 and Limitation 2. Further elaborate on the motivation and theoretical foundation of the method.
>
> Our key motivation is that reasoning difficulty is highly non-uniform within a single chain of thought: only a few steps are genuinely difficult, while many others are routine. Applying deep reasoning uniformly therefore creates unnecessary redundancy, which motivates **switching modes within a response** rather than only at the problem level.
>
> The key question is then when to switch. Prior work [1, 2] has shown that **reasoning difficulty and quality can be predicted from signals derived from a model’s internal token distribution**. Building on this, we use token confidence as a proxy for local reasoning difficulty.
>
> We further adopt a sliding-window formulation because single-token confidence is often noisy and may fail to capture semantically meaningful difficult regions[3, 4]. In contrast, **segment-level confidence provides a more robust switching signal** and better reflects the role of intermediate reasoning as semantic progress rather than isolated token choices, as illusatraed in Figure 1.
>
> **Sliding-window confidnce is not purely heuristic.** As discussed in Appendix C, the confidence score has a principled interpretation as a measure of distributional concentration, and our experiments show that it works better in practice than related alternatives such as entropy, max probability, and margin.
>
> Together, these points provide both the motivation and the theoretical basis for the method.
>
> **Reference**
>
> [1] Fu et al. Deep Think with Confidence.
>
> [2] Kang et al. Scalable best-of-n selection for large language models via self-certainty.
>
> [3] Pan et al. Specreason: Fast and accurate inference-time compute via speculative decoding.
>
> [4] Li et al. What makes reasoning models different? follow the reasoning leader for efficient decoding.
>
> ---
> > Limitation 1 and Weakness 2. More downstream evaluation tasks and models.
>
> We additionally include code-generation and commonsense benchmarks. We also expand the model set with three additional base models: DeepSeek-R1-Distill-Qwen-1.5B, DeepSeek-R1-Distill-Llama-8B/70B, with the latter two extending to the Llama family.
>
> With these additions, our evaluation now covers **8 datasets** spanning math, STEM, code generation, and commonsense reasoning, and 7 models from **4 model families**, covering a broad range of scales **(1.5B/7B/8B/14B/32B/70B)**.
>
> ||Humaneval||LiveCodeBench||StrategyQA||
> |-|-|-|-|-|-|-|
> |**Method**|**Pass@1**|**#Token**|**Pass@1**|**#Token** |**Pass@1**|**#Token**|
> |**Qwen3-8B**|91.95|2881|44.34|8352|77.32|1304|
> |**MixReasoning**|93.29|2643|45.00|7689|77.04|1184|
>
> ||Math500||AIME24||
> |-|-|-|-|-|
> |**Method**|**Pass@1**|**#Token**|**Pass@1**|**#Token** |**Pass@1**|**#Token**|**Pass@1**
> |**DeepSeek-R1-Distill-Qwen-1.5B**|83.45|4896|66.67|7130|||||
> |**MixReasoning**|83.65|4091|65.98|6034||
> |**DeepSeek-R1-Distill-Llama-8B**|86.35|3896|45.56|11116|||||||
> |**MixReasoning**|87.60|3039|50.00|10391|
> |**DeepSeek-R1-Distill-Llama-70B**|93.07|2851|71.20|7836|
> |**MixReasoning**|93.60|2317|70.00|6396|
>
> ---
> > Part 1 of Limitation 3. Discuss the potential real-world application scenarios of the method.
>
> We thank the reviewer for this suggestion. We believe MixReasoning has clear practical value in latency-sensitive LLM applications, such as chat assistants, tutoring systems, coding copilots, and agentic workflows. In these settings, long reasoning traces often increase latency and serving cost, and excessively verbose traces can also reduce readability for end users. By allocating deeper reasoning only to difficult substeps, MixReasoning can reduce waiting time and compute overhead while producing answers that are more concise and easier to read.
>
> > Weakness 3 and Part 2 of Limitation 3. Discuss further optimization, and whether it can assist the training process.
>
> We thank the reviewer for this question. We believe that there is meaningful room for further work. In the current paper, the switching controller is not learned end-to-end. A natural extension is to replace it with a learned policy that directly controls reasoning depth during generation, for example via RL or imitation learning. In particular, our sliding-window confidence could be incorporated as a fine-grained reward signal, providing step-level supervision for reasoning depth rather than relying solely on the final-answer reward. Moreover, the responses produced by MixReasoning are difficulty-aware mixtures of concise and detailed reasoning, which could serve as useful supervision for distillation or SFT.
>
> ---
> Thanks again for your feedback. If you have any remaining concerns, please let us know so that we can clarify further.

---

> > ### Author Rebuttal · Reviewer_YdxE · 2026-04-02
> >
> > Thank you for the authors’ response. I have the following additional questions.
> >
> > + The improvement in model performance is not significant. On HumanEval and StrategyQA, the gain is only around 100 tokens, which may be due to variance. The authors should provide statistical measures such as standard deviation to further substantiate the validity of the experimental results.
> >
> > + MixReasoning requires training a Concise LoRA module. What is the additional training overhead incurred?
> >
> > + Frequent switching of token-level reasoning modes may lead to instability and reduced readability in the generated content.

---

> > > ### Author Response · Authors · 2026-04-02
> > >
> > > Thank you for the further discussion. We appreciate these additional questions and provide additional empirical evidence and clarification below.
> > >
> > > > Additional Question 1. The improvement in model performance is not significant on Humaneval and StrategyQA.
> > >
> > > Thank you for the follow-up. We now report **mean $\pm$ std** over 5 independent runs, and additionally show the absolute and relative token reduction. The reductions are **consistently larger than the run-to-run variation** (83 vs. 238 and 23 vs. 120), suggesting that the token savings is **not to be explained by variance**.
> > >
> > > To further contextualize the results, we also reproduced two representative CoT-compression baselines [1, 2]. The results show that MixReasoning is **competitive in token compression while largely maintaining the performance, whereas other baselines incur a clear accuracy drop**.
> > >
> > > ||Humaneval|||StrategyQA|||
> > > |-|-|-|-|-|-|-|
> > > |**Method**|**Pass@1**|**#Token**|$\Delta$|**Pass@1**|**#Token**|**$\Delta$**|
> > > |**Qwen3-8B**|91.95 $\pm$ 1.65|2881 $\pm$ 71|-|**77.32 $\pm$ 0.56**|1304 $\pm$ 15|-|
> > > |**BeConcise [1]**|92.31 $\pm$ 1.89|2842 $\pm$ 91|-39 (-1.4%)| 76.31 $\pm$ 0.81|1274 $\pm$ 19|-30 (-2.3%)
> > > |**Deer [2]**|91.43 $\pm$ 2.43|2627 $\pm$ 128| **-254 (-8.8%)**|75.79 $\pm$ 0.94|1229 $\pm$ 27|-75 (-5.8%)|
> > > |**MixReasoning**|**93.29 $\pm$ 1.24**|2643 $\pm$ 83|-238 (-8.3%)|77.04 $\pm$ 0.75|1184 $\pm$ 23|**-120 (-9.2%)**|
> > >
> > > We would also like to clarify **why the token reduction on Humaneval and StrategyQA is less “pronounced”** than on math-heavy benchmarks:
> > > - Their responses are already short (around 1k–2k tokens), and a larger fraction of the output is task-essential rather than redundant, such as generated code in HumanEval and supporting implicit reasoning in StrategyQA.
> > > - This limitation is not unique to our method, **other efficient reasoning baselines exhibit the same phenomenon as well**.
> > >
> > > **Reference**
> > >
> > > [1] Renze et al. The benefits of a concise chain of thought on problem-solving in large language models.
> > >
> > > [2] Yang et al. Dynamic early exit in reasoning models. (ICLR 2026)
> > >
> > > ---
> > > > Additional Question 2. Additional training overhead incurred by concise LoRA module.
> > >
> > > The additional training overhead is minimal. As described in Sec. 4.1, we train the concise LoRA on a single **~40K-example** dataset, with trainable parameters accounting for only about **1% of the base model**. In our experiments, training the LoRA for Qwen3-8B takes **less than 1 hour** on 4 H100 GPUs.
> > >
> > > ---
> > > > Additional Question 3. Frequent switching lead to instability and reduced readability in the generated content.
> > >
> > > Thank you for raising this concern. We would like to clarify that switching in MixReasoning does not introduce abrupt state inconsistency and instability:
> > >
> > > - **The KV cache is mode-specific**, and switching reuses the KV cache of the destination mode rather than mixing KV caches or activations. So the generation process within each mode remains internally consistent.
> > > - Concise mode and base mode are **still the same underlying model**, with the LoRA adapter mainly controlling reasoning detail rather than inducing a fundamentally different reasoning behavior.
> > > - Token confidence is measured at the segment level based on sliding-window confidence, and **a dwell constraint prevents rapid oscillations** by enforcing at least one window of decoding before another switch.
> > >
> > > For readability, we find that MixReasoning improves rather than degrades the generated content relative to the base model, as it **reduces excessively verbose traces and redundant self-reflection**, which are often less consistent with human reasoning patterns. A qualitative example is shown in Fig. 7 of the Appendix, and we further add more qualitative examples at [anonymous link](https://anonymous.4open.science/r/icml-rebuttal-1866-4EBD/README.md) to strengthen this evidence.
> > >
> > > Overall, both the **mechanism design** and the **qualitative evidence** suggest that MixReasoning does not lead to substantial instability or reduced readability in practice.

---

### Official Review · Reviewer_cGmq · 2026-02-26

**Soundness:** 3
**Presentation:** 4
**Significance:** 3
**Originality:** 3
**Overall Recommendation:** 5
**Confidence:** 4

**Summary:**

This paper introduces MixReasoning, a framework designed to optimize Reasoning Models by dynamically adjusting their reasoning effort within a single response. Not all steps in a complex problem require the same level of deliberation, therefore the authors propose a method to switch between an intensive "Thinking Mode" and a "Concise Mode" based on task difficulty. Technically, this is achieved by using a LoRA adapter as a controllable "task vector" alongside a sliding-window token confidence metric that identifies when the model is at a "pivotal" versus "routine" step. Empirical results across multiple benchmarks demonstrate that MixReasoning significantly reduces token usagae—cutting reasoning lengths by 13-49%—while maintaining or even slightly improving overall accuracy on benchmarks such as MATH, GSM8K, AIME24, GPQA, and CommonsenseQA.

**Compliance With Llm Reviewing Policy:**

Affirmed.

**Final Justification:**

As I mentioned in my rebuttal acknowledgement, I believe the method is promising and the results are good enough to warrant acceptance. Efficient chain of thought has recently become an important field and this paper is a valuable addition.

**Key Questions For Authors:**

1) From my understanding, there seems to be a distribution mismatch regarding how the LoRA adapters are trained vs how they are used - a) During training, they are trained to produce the final solution (no thinking) but at inference they are used during thinking to produce concise spans of thinking traces. b) Additionally, during training the model along with LoRA adapter is used for processing the whole sequence which means that all the activations are produced as a result of applying the LoRA adapters but during inference if the model decides to switch to concise thinking the the sequence generated upto the previous window has keys and values not generates via the LoRA adapters. Do these two mismatches cause any issues?

**Limitations:**

yes

**Strengths And Weaknesses:**

Strengths

1. The paper tackles an important problem of overthinking in reasoning models. Its technical contributions are sound and claims are validated with sufficient experiments.
2. I believe this is the first paper which proposes switching within a response. Most existing work on routing or efficient reasoning treats a problem as a single unit i.e. either a small model is used for the whole thing or a large reasoning model for the whole thing.
3. The presentation is clear and the paper is well written and well structured
4. The empirical results span a range of models and datasets which makes the experiments thorough.

Weaknesses
1. While the experimental breadth is thorough, the authors only report the mean values for Pass@1 and token counts. The lack of standard deviation or error bars in the tables makes it difficult to assess the statistical significance of the reported performance gains, especially in cases where the margins between MixReasoning and the baseline models are narrow
2. It would be good to see more evidence regarding how the model switches between concise and thinking modes: a) It would be nice to report how many percentage of tokens are output by each mode for each dataset across all questions. Ideally, we would see that AIME/GPQA use more thinking mode tokens compared to concise and the trend would be opposite for GSM8K and MATH. b) The qualitative example in the appendix only covers gsm8k I think? it would be nice to see examples from AIME and GPQA as well

---

> ### Author Rebuttal · Authors · 2026-03-30
>
> We thank the reviewer for their thoughtful review and are glad that you found our work to address an “important problem,” with “sound” technical contributions and “thorough” experiments. We respond to the critiques and comments below.
>
> ---
> > Weakness 1. While the experimental breadth is thorough, the authors only report the mean values for Pass@1 and token counts. The lack of standard deviation or error bars in the tables makes it difficult to assess the statistical significance of the reported performance gains.
>
> Thank you for this helpful suggestion. Below we report the standard deviation for both Pass@1 and generated token counts over the five independent runs. We also clarify that our main claim is not that MixReasoning significantly improves accuracy over the base model, but that it reduces CoT cost while largely preserving performance across problems of different difficulty. In some cases, MixReasoning indeed brings performance improvements, which we believe arise because MixReasoning **avoids unnecessarily verbose reasoning** and **reduces the chance of drifting into incorrect reasoning paths**, as illustrated in Figure 7 of the Appendix.
>
> ||AIME24||GPQA||MATH500||GSM8K||
> |-|-|-|-|-|-|-|-|-|
> |**Method**|**Pass@1**|**#Token**|**Pass@1**|**#Token** |**Pass@1**|**#Token**|**Pass@1**|**#Token**|
> |**Qwen3-8B**|64.67 $\pm$ 4.04|11975 $\pm$ 404|57.32 $\pm$ 0.97|8642 $\pm$ 80|92.30 $\pm$ 0.60 |5104 $\pm$ 57 |95.45 $\pm$ 0.29 | 2403 $\pm$ 17|
> |**Concise Mode**|30.00 $\pm$ 7.63|6021 $\pm$ 763|45.56 $\pm$ 3.24|5173 $\pm$ 123 |82.40 $\pm$ 0.31|1535 $\pm$ 114 |92.89 $\pm$ 0.44 | 534 $\pm$ 12
> |**MixReasoning**|67.33 $\pm$ 5.51|8873 $\pm$ 551|57.17 $\pm$ 1.32 | 6734 $\pm$ 92|92.76 $\pm$ 0.77| 3483 $\pm$ 47|95.88 $\pm$ 0.23 | 1389 $\pm$ 26
>
> ---
> > Weakness 2. It would be good to see more evidence regarding how the model switches between concise and thinking modes. (a) Thinking mode percentage across different datasets. (b) More qualitative examples from GPQA and AIME.
>
> **(a)** Thank you for this valuable suggestion. We agree that this analysis would strengthen the empirical support for our switching mechanism. Below, we report the **thinking-mode token ratio** and the total **number of mode switches** on each dataset use Qwen3-8B. We find that AIME24 and GPQA use a higher proportion of thinking-mode tokens and more swicthes than MATH500 and GSM8K, **consistent with their higher difficulty**.
>
> | |AIME24|GPQA|Math500|GSM8K|
> |-|-|-|-|-|
> | **Thinking mode percentage** (%) |53.82|40.83|31.17|28.93|
> | **Switching nums** |66|38|21|13|
>
> **(b)** Thank you for this valuable suggestion. We agree that examples from AIME and GPQA would further strengthen the qualitative evidence for our switching mechanism. We will incorporate more examples in the revised version.
>
> ---
> > Part 1 of Question 1. Training-target and inference-usage mismatch for the concise LoRA adapter.
>
> Thank you for raising this point. We clarify that the concise LoRA adapter is not trained on answer-only targets: as defined in Eq. (1), the supervision includes both a short explanation $t_{1:m}$ together with the final answer $a$. In practice, these targets are generated by the instruct (i.e., non-reasoning / pre-RL) model and therefore still contain basic problem analysis, decomposition, and solution steps; compared with long-form reasoning traces, they mainly differ in omitting overly verbose self-correction and self-reflection.
>
> Accordingly, as discussed in sec. 3.1, the concise adapter $\Delta\theta$ is better interpreted as a **task vector** that primarily controls CoT length, rather than as inducing a fundamentally different reasoning behavior. In this sense, the concise adapter steers the model toward a **more compact reasoning style** while preserving the base model’s **underlying reasoning ability**. The empirical results in Table 1 are consistent with this view: larger $\alpha$ leads to progressively shorter generations.
>
> ---
> > Part 2 of Question 1. Activation / KV-cache mismatch under inference-time mode switching.
>
> Thank you for raising this important concern. We believe this concern may stem from a slight misunderstanding of our implementation details. In our method, we do not continue decoding under a mixed KV cache. As described in Sec. 3.3 and Appendix C.3, the KV cache is explicitly mode-dependent, since different LoRA strengths produce different activations. When switching from thinking mode to concise mode, we reuse the KV cache already built for concise mode on the shared prefix, and only prefill the new tokens generated in another mode. Hence, at each decoding step, the active cache is internally consistent with the current mode; we are not directly reusing non-LoRA KV states while decoding with the LoRA adapter or vice versa.
>
> ---
> Thanks again for the thoughtful and constructive feedback. If you have any remaining concerns, we would be happy to further clarify.

---

> > ### Author Rebuttal · Reviewer_cGmq · 2026-04-03
> >
> > I believe my concern was about the error bars which has been addressed. I think this method is promising and achieves good results for efficient chain of thought. I will raise my score to accept.

---

> > > ### Author Response · Authors · 2026-04-03
> > >
> > > Thank you very much for the update and for raising your score to accept. We sincerely appreciate your thoughtful feedback and are glad that the concern about the error bars has been addressed. We are encouraged that you find the method promising and effective for efficient chain-of-thought reasoning.

---

### Official Review · Reviewer_uRdy · 2026-03-12

**Soundness:** 3
**Presentation:** 3
**Significance:** 2
**Originality:** 2
**Overall Recommendation:** 3
**Confidence:** 3

**Summary:**

This paper proposes MixReasoning, a framework that dynamically switches between detailed and concise reasoning modes within the same response based on local CoT step difficulty. A concise LoRA adapter is trained via SFT on short correct solutions from non-thinking models, and its strength alpha is scaled at inference time to interpolate between full reasoning (alpha=0) and concise output. This paper also uses sliding-window mechanism to determine when reasoning is routine versus challenging. Experiments on several math reasoning benchmarks show the effectiveness of token reduction while maintaining the reasoning accuracy.

**Compliance With Llm Reviewing Policy:**

Affirmed.

**Final Justification:**

After considering both the original paper and the rebuttal, I would like to keep my assessment.

My main concern remains unresolved: the claimed efficiency gains are not yet sufficiently convincing in practical inference settings. Since the method relies on frequent mode switching, rollback, and mode-dependent KV-cache handling, reduced token counts alone are not enough to establish real end-to-end efficiency improvements. While the rebuttal provided clarifications, it did not fully address the potential runtime and systems overhead introduced by switching, which is critical to the claimed contribution.

To the second round rebuttal:

I want to remind the authors that single sample latency may not comprehensively measure the cost due to high variance. Such frequent mode switch will systematically destroy KV cache and modern LLM inference system (e.g., vLLM), which will likely degrade the overall throughout.

**Key Questions For Authors:**

Please see the weakness.

**Limitations:**

yes

**Strengths And Weaknesses:**

### Strengths

1. The core observation that reasoning difficulty varies at the sub-step level within the same response is interesting and supported example in Figure 3.
2. The LoRA strength scaling mechanism is practical and clean, and compatible with batch inference. Though I do think integrate these mechanisms can be hard to implement in modern inference engines such as vLLM and SGLang.  Could the authors provide more details on how they implemented in vLLM?
3. The experimental evaluation covers different model scales and math benchmarks, demonstrating the effectiveness of the proposed method in token reduction while maintaining reasoning performance.

### Weaknesses

1. The accuracy differences between MixReasoning and the base model are small, I wonder whether MixReasoning can increase the strength to improve the reasoning effort or can only be used for token reduction. Also, I suggest the authors to test on AIME25 since AIME24 may be leaked for the adopted models.
2. Table 4 shows that the concise adapter alone performs very poorly on AIME24 (26.66–30.00 vs 64.67 ORI.), indicating that the concise mode cannot handle hard problems, which limits the token savings achievable on hard problems. Could the authors report the fraction of tokens generated in different modes?
3. The token confidence is presented as a proxy for step difficulty, but this paper does not provide quantitative validation of this claim beyond the visualization. A controlled experiment correlating confidence with step-level error rates would strengthen the claim.
4. The rollback mechanism regenerates one window of tokens when switching from concise to thinking mode, but this component should be ablated studied. It is unclear how much accuracy is preserved by rollback versus simply continuing in thinking mode from the current position. Also, how to manage the KV cache during rollback?
5. The controller introduces many hyperparameters which interact with each other, which may not be robust across different settings.

---

> ### Author Rebuttal · Authors · 2026-03-30
>
> We thank the reviewer for the thoughtful feedback. We respond to critiques and comments below.
>
> ---
> > Strength 1. Could the authors provide more details on how they implemented in vLLM?
>
> We thank the reviewer for this practical question. Our vLLM implementation is lightweight: MixReasoning uses only two modes in practice, thinking mode ($\alpha$=0) and concise mode, which can be served either as one base model with a LoRA adapter toggled on/off or as two workers. We extend vLLM to track sliding-window confidence online from token log-probabilities and check the switching condition after each decoding step in the output processor. When triggered, decoding stops in the current mode and resumes in the other mode from the current prompt and generated prefix. Because vLLM already supports prefix/KV-cache reuse for both the base and LoRA paths, the switching overhead is small and remains compatible with batched inference. The main changes are in `vllm/v1/engine/logprobs.py` and `vllm/v1/engine/output_processor.py`.
>
> ---
> > Part 1 of Weakness 1. Wondering whether MixReasoning can increase the strength to improve the reasoning effort or can only be used for token reduction.
>
> We would like to clarify that our main claim is not that MixReasoning substantially improves accuracy over the base model, but that it **reduces CoT cost while largely preserving performance across problems of different difficulty**. In some cases, we do observe **performance gains**, which we believe arise because MixReasoning avoids unnecessarily verbose reasoning and reduces the chance of drifting into incorrect reasoning paths. A qualitative example illustrating this behavior is provided in Figure 7 of the Appendix.
>
> ---
> > Part 2 of Weakness 1. I suggest the authors to test on AIME25 since AIME24 may be leaked for the adopted models.
>
> Thanks for the suggestion. We conduct a minimal experiments on AIME25 below. We will include full experiments in the revised version.
>
> ||AIME25||
> |-|-|-|
> |**Method**|**Pass@1**|**#Token**|
> |**Qwen3-8B**|57.00|12365|
> |**Concise Mode**|50.00|9838|
> |**MixReasoning**|57.67| 11320 |
>
> ---
> > Weakness 2. Could the authors report the fraction of tokens generated in different modes?
>
> We report the thinking-mode token ratio and the total number of mode switches on each dataset use Qwen3-8B. We find that AIME24 and GPQA use a higher proportion of thinking-mode tokens and more swicthes than MATH500 and GSM8K, consistent with their higher difficulty.
>
> | |AIME24|GPQA|Math500|GSM8K|
> |-|-|-|-|-|
> | **Thinking mode percentage** (%) |53.82|40.83|31.17|28.93|
> | **Switching nums** |66|38|21|13|
>
> ---
> > Weakness 3. The token confidence is presented as a proxy for step difficulty. A controlled experiment correlating confidence with step-level error rates would strengthen the claim.
>
> We agree that a quantitative validation would strengthen this claim. We conducted a minimal controlled experiment: we sampled 20 questions, generated 3 responses per question, and computed the confidence and step-level error rate for each sub-step. The results below show a clear correlation.
>
> ||$C \le 15$ | $15 \lt C \le 18$ | $18 \lt C \le 20$ | $20 \le C$ |
> |-|-|-|-|-|
> |**Error rates** (%)| 33 | 18 | 16 | 7 |
>
> ---
> > Weakness 4. Ablation study on rollback machanism.
>
> We agree that the rollback mechanism is an important part of our method. As shown below, removing rollback causes performance drops. This supports our claim that the early tokens of a reasoning step can strongly shape the subsequent reasoning trajectory, and that rollback is crucial for correcting difficult segments after a mode switch.
>
> ||AIME24||MATH500||
> |-|-|-|-|-|
> |**Method**|**Pass@1**|**#Token**|**Pass@1**|**#Token** |**Pass@1**|**#Token**|**Pass@1**
> |**Qwen3-8B**|64.67|11975|92.30|5104|
> |**MixReasoning w/o rollback**|60.17|8346|91.65|3578
> |**MixReasoning w/ rollback**|67.33|8873|92.76|3483|
>
> **KV cache managment**. During rollback, we keep the KV cache of the shared prefix unchanged, discard only the KV states of the rolled-back span. Thus, the overhead scales only with the switched span rather than the full prefix.
>
> ---
> > Weakness 5. The controller introduces many hyperparameters which interact with each other, which may not be robust across different settings.
>
> Thanks for raising this important concern. We believe there may be a slight misunderstanding about the number and role of hyperparameters in our method. The mode controller introduces only two hyperparameters: the sliding-window size and the switching thresholds. For the window size, our ablation over 8, 16, and 32 shows that performance is **not sensitive** as long as the window is not too small or too large, so we fix it to 16 in all experiments. For the thresholds, the sliding-window confidence distributions are highly **consistent across benchmarks and model variants** (Figure 4), suggesting that the thresholds are transferable and do not require heavy per-dataset or per-model tuning.

---

> > ### Author Rebuttal · Reviewer_uRdy · 2026-04-03
> >
> > Thank you for your response.
> >
> > The new AIME25 results show that the performance gain is only marginal (or some variance), while the reported mode-switch statistics suggest that switching occurs very frequently. This raises a critical concern: such frequent switching is likely to introduce nontrivial inference overhead, which may offset the claimed efficiency gains in token reduction. To make the efficiency claim convincing, the paper should report actual wall-clock latency, throughput, or other end-to-end inference efficiency metrics, rather than token counts alone. At present, it remains unclear whether MixReasoning yields meaningful practical savings.
> >
> > In addition, I remain unconvinced by the explanation of KV-cache handling under LoRA switching. Different LoRA adapters modify the hidden-state transformation, so the cached KV states are generally adapter-dependent. If the method reuses KV states computed under one LoRA setting after switching to another, the correctness of subsequent decoding is unclear and needs to be justified much more carefully. On the other hand, if the KV states must be recomputed after each switch, then the overhead could be substantial. The same concern applies to the rollback mechanism: even if only part of the cache is discarded and recomputed, the actual runtime cost may still be significant when switching happens frequently.

---

> > > ### Author Response · Authors · 2026-04-03
> > >
> > > Thank you for the further discussion. We appreciate these additional questions, as they help us clarify several important aspects of our method. Below, we provide further clarification and additional empirical evidence regarding these concerns.
> > >
> > > > Part 1 of additional question 1. The new AIME25 results show that the performance gain is only marginal (or some variance).
> > >
> > > We now report **mean ± std** over 5 independent runs and include both absolute and relative token reduction. We believe MixReasoning remains competitive: compared with Qwen3-8B, it improves Pass@1 from **57.00 to 57.67** while reducing generation length by **1,045 tokens (-8.5%)**. This reduction is substantially larger than the run-to-run variation and is therefore **not to be explained by variance**.
> > >
> > > To further contextualize the results, we also reproduced two representative CoT-compression baselines [1, 2]. Both achieve only marginal token compression (**-1.1% and -2.3%**, respectively) and also show **lower accuracy**, whereas MixReasoning delivers a **much stronger efficiency gain while slightly improving performance**. We therefore believe the AIME25 results support that MixReasoning can achieve a competitive and favorable accuracy-efficiency trade-off compared with existing baselines.
> > >
> > > ||AIME25|||
> > > |-|-|-|-|
> > > |**Method**|**Pass@1**|**#Token**|**$\Delta$**|
> > > |**Qwen3-8B**|57.00 $\pm 3.67$|12365 $\pm$ 384|-|
> > > |**Beconcise [1]**|53.33 $\pm 4.35$|12229 $\pm$ 412|-136 (-1.1%)|
> > > |**Deer [2]**|55.00 $\pm$ 3.33|12087 $\pm$ 434|-278 (-2.3%)|
> > > |**MixReasoning**|**57.67 $\pm$ 4.16**|**11320 $\pm$ 309**|**-1045 (-8.5%)**|
> > >
> > > [1] Renze et al. The benefits of a concise chain of thought on problem-solving in large language models.
> > >
> > > [2] Yang et al. Dynamic early exit in reasoning models. (ICLR 2026)
> > >
> > > ---
> > > > Part 2 of additional question 1. Frequent switching is likely to introduce nontrivial inference overhead which may offset the claimed efficiency gains in token reduction. Meaningful practical savings evidance.
> > >
> > > Thank you for this concern. We respectfully note that **the paper already reports end-to-end latency in the main text** rather than relying on token counts alone. Specifically, **Sec. 4.3 and Fig. 6** present the end-to-end latency of Qwen3-8B on various benchmarks, measured with vLLM. These results show that the reduction in generated tokens is not offset by switching overhead, and that **MixReasoning consistently reduces end-to-end latency** relative to the base model across all benchmarks.
> > >
> > > To further strengthen this point, we now additionally report AIME25 end-to-end latency, where MixReasoning **still shows practical speedup**. We therefore believe the paper already provides **direct evidence that MixReasoning yields meaningful practical savings**.
> > >
> > > ||AIME25|
> > > |-|-|
> > > |**Method**|**Latency (s)**|
> > > |**Qwen3-8B**|15.64|
> > > |**MixReasoning**|14.06|-
> > >
> > > ---
> > > > Part 1 of additional question 2. KV-cache handling mechanism.
> > >
> > >
> > > Thank you for pointing this out. In our method, we **do not continue decoding under a mixed KV cache**. As described in Sec. 3.3 and Appendix C.3, the KV cache is explicitly mode-dependent: **each mode maintains its own cache**, and we never decode with a mixed KV cache. When switching modes, we reuse the cache already built for the destination mode on the shared prefix, and only prefill the tokens generated after divergence. So the generation process within each mode remains internally consistent.
> > >
> > > And we would also like to clarify prefilling is highly parallelizable, so the overhead is minimal as discussed in Sec. 3.3 (even a long prefill often takes roughly the wall-clock time of generating **only 1 autoregressive token**). So, the switching will not offset the claimed efficiency gains as shown in the end-to-end latency comparison in Sec. 4.3 and Fig. 6.
> > >
> > > > Part 2 of additional question 2. Rollback mechanism overhead could be substantial.
> > >
> > > Regarding rollback, its cost is **already included in our token accounting**. Moreover, we believe switching is **not highly frequent in practice**: on AIME24, the model generates about **10k** tokens in total, while switches occur only **66** times; with a rollback window of 16 tokens, this corresponds to at most 528 re-generated tokens, i.e., **about 5% of the total**. On GSM8K, switching happens **only 13** times. We therefore do not view the rollback overhead as substantial in our setting.

---

### Official Review · Reviewer_zPQD · 2026-03-15

**Soundness:** 3
**Presentation:** 3
**Significance:** 2
**Originality:** 2
**Overall Recommendation:** 4
**Confidence:** 3

**Summary:**

Reasoning by CoT tends to be chatty with talking while the performance is improved. At the moment of speaking with inertia, the authors paid attention. When speaking with inertia, the author hypothesize that the confidence is already high enough and not even to say it, and when the value is high, the provided model switches to concise mode, and when the confidence is low, the model is actively talking about the thought process, and the model from the perspective of deepening its thinking by actively talking about the thought process. Then, the authors measured this confidence and performed LoRA tuning to intervene in the thought mode. As a result, this study succeeded in reducing the loss of performance and limiting the output token to about seventy percent.

**Compliance With Llm Reviewing Policy:**

Affirmed.

**Final Justification:**

I showed the following majour 2 concerns.
- Questions whether the experimental setup is sufficiently sound and comparative (Pass@1, mostly math benchmarks, and limited trade-off analysis.)
-  if it is high trust for the LLM, why can the reconsideration and revision be clearly reduced?
The first concern is solved at the first rebuttal.
The second concern is almost solved by the last rebuttal.
Thus, I should update overall score.

**Key Questions For Authors:**

1. Doubts about the idea level: When you strongly believe that the model is subjectively correct, confidence is raised, and confidence does not rise when it is truly objectively correct, regardless of the model. So, if the model is wrong once, don’t you push forward with a short reasoning, with a strong belief in your mistake? Does that mean you’re not even taking the opportunity to doubt your thoughts?
2. Doubts about the evaluation metrics: When focusing on cost and speed, we evaluate that the setting that generates only one time with a short token, that is, the pass@1 / token length indicator in this paper is useful. However, as long as there are doubts about the above, if it is an academic paper rather than a product advertisement, I think that the performance evaluation at pass@k ($2 \le k \le 5 $) should be clarified and the nature and trade-off of this method.
3. Question to deliverables: Will the model, which is a workaholic that tuned the length of this thought, only work for the trained domain? If you look at Confidence and switches the length of your thoughts, you will be effective not in task-dependent, but at any other domain, but in any other domain, but please tell us why you limited it to math tasks.
4. Average is the only analysis of the length of the token, but in this case, if there is a long Reassoning or if there is no valid Decision pivot, the length of the CoT is reduced to zero, so the value will be shaken greatly, and the effect of CoT shortening cannot be judged correctly.
Please tell me about the analysis of the CoT length of the model in the previous study and the proposed method, and what is the statistical value of 1/4, median, 3/4, and variance.

**Limitations:**

yes

**Strengths And Weaknesses:**

Strengths
- In all experiments, the score was very high, and at first glance it was a development of a theory that there was no gap.
- On the other hand, although it is a simple expression development in appendix, it shows that kl divergence etc. are maintained, and in the theoretical part it was likable.

Weaknesses
- There was no mention of trade-offs, only pass@1, only STEM tasks centered on mathematics, and metrics of experiments were selected, and it was advertised rather than a comparative experiment, and I did not feel soundness in the experimental method.

- This study has improved in terms of eliminating waste in the thought process, but it does not improve or improve the wisdom and shortcomings that are the ultimate goal of the entire field. In particular, there is a concern that shortening tokens is losing the goodness of machine learning that essentially reaches out of distribution

---

> ### Author Rebuttal · Authors · 2026-03-31
>
> Thanks for the feedbacks. We respond to critiques and comments below.
>
> ---
> > Weakness 1 and Question 3. Questions whether the experimental setup is sufficiently sound and comparative (Pass@1, mostly math benchmarks, and limited trade-off analysis.)
>
> Thank you for the comment. We respectfully disagree that the metrics are selectively chosen or non-comparative. Our evaluation follows the standard protocol widely adopted in efficient reasoning work: we report both Pass@1 and average generated tokens, as these metrics jointly capture the central accuracy–efficiency trade-off studied in this line of research. This setup is also consistent with prior work [1,2,3,4,5], ensuring a fair and directly comparable assessment. We also report end-to-end latency and discuss switching overhead, so trade-offs are explicitly analyzed rather than omitted.
>
> Regarding downstream task diversity, we chose the current benchmarks because they are widely used reasoning benchmarks in this literature, and we have also additionally included CommonsenseQA in the appendix to demonstrate generalization beyond STEM tasks. To further strengthen the evidence, we report the results of commonsense and code-generation benchmarks below.
>
> ||Humaneval||LiveCodeBench||StrategyQA||CommonsenseQA||
> |-|-|-|-|-|-|-|-|-|
> |**Method**|Pass@1|#Token|Pass@1|#Token|Pass@1|#Token|Pass@1|#Token
> |**Qwen3-8B**|91.95|2881|44.34|8352|77.32|1304|83.39|1183
> |**MixReasoning**|93.29|2643|45.00|7689|77.04|1184|83.90|895
>
> **Reference**
>
> [1] Fang et al. Thinkless: LLM leans when to think
>
> [2] Ma et al.  CoT-Valve: length-compressible chain-of-thought tuning
>
> [3] Fu et al. Deep think with confidence
>
> [4] SpecReason: Fast and Accurate Inference-time compute via
> speculative reasoning
>
> [5] Luo et al. O1-Pruner: Length-harmonizing fine-tuning for o1-like reasoning pruning
>
> ---
> > Weakness 2. This study has improved reasoning efficiency, but it does not improve the wisdom and shortcomings that are the ultimate goal of the entire field. There is a concern that shortening tokens is losing the goodness of machine learning that essentially reaches out of distribution
>
> Thank you for this thoughtful comment. We respectfully disagree that improving reasoning efficiency is separate from improving the overall quality of reasoning systems. Excessive reasoning length are important shortcomings of current LRMs: they increase latency and serving cost, reduce readability, and deviate from human-like reasoning patterns.
>
> We believe that naively shortening reasoning traces can be harmful, if it uniformly suppresses deliberation on difficult steps. In fact, this is exactly one of the motivations of our paper: global compression may truncate pivotal reasoning, induce underthinking, and hurt the performance. Our method is designed to avoid this failure mode. MixReasoning does not compress reasoning uniformly; instead, it switches between concise and detailed reasoning. Thus, the goal is not to reduce “intelligence,” but to allocate compute more effectively.
>
> ---
> > Question 1. Idea-level concern: Confidence may not align with objective correctness, the proposed switching strategy could reinforce confident mistakes.
>
> We agree that confidence is not equivalent to objective correctness and this is not the assumption made by our method.
> - Concise mode does not mean “stop thinking”: Our method is designed to switch between concise and detailed reasoning dynamically rather than a one-way truncation of thought, analogous to human fast and slow thinking in dual-process theories.
> - Still allows self-correction/reflection: Reasoning is treated as a dynamic process, once a low-confidence segment is detected, it returns to detailed reasoning, explicitly creating room for reconsideration and correction.
>
> ---
> > Question 2. I think that the performance evaluation at pass@k (2-5) should be clarified.
>
> We agree that Pass@k is a useful supplementary metric. We would like to clarify that our method is not primarily aimed at test-time scaling through more sampled trajectories. Instead, MixReasoning improves the efficiency of each individual reasoning path. For this reason, Pass@1 is the natural primary metric for our method. We additionally report Pass@2–5 as below.
>
> ||AIME24||||Math500||||
> |-|-|-|-|-|-|-|-|-|
> |**Method**|Pass@2|Pass@3|Pass@4|Pass@5|Pass@2|Pass@3|Pass@4|Pass@5|
> |**Qwen3-8B**|71.33|72.00|72.67|73.33|93.60|94.50|94.65|95.12|
> |**MixReasoning**|73.67|76.67|78.67|80.00|94.12|94.66|94.88|95.00|
>
> ---
> > Question 4. Mean token length alone is insufficient, and asks for a fuller statistical characterization.
>
> Thanks for this comment. We additionally report the standard deviation, median, 1/4, and 3/4 of CoT length, computed over the full set of examples in each benchmark.
>
> ||AIME24 (#Token)||||Math500 (#Token)||||
> |-|-|-|-|-|-|-|-|-|
> |**Method**|Std|1/4|Median|3/4|Std|1/4|Median|3/4|
> |**Qwen3-8B**|4660|7729|10785|15856|3970|1989|3269|5909|
> |**MixReasoning**|4773|6016|9409|14720|3704|1503|2441|4578|

---

> > ### Author Rebuttal · Reviewer_zPQD · 2026-04-06
> >
> > I understood that Pass@1 is a standard that follows the previous study. We also confirmed that HumanEval, LiveCodeBench, and StrategyQA CommonsenseQA have improved performance while maintaining scores. Therefore, we believe that the weakness one and question three have been resolved.
> >
> > I object to the idea of human-like reasoning and improving wisdom  as a trivial-relationship. Human-like reasoning is just one example of the way of thinking. On the other hand, I understood that it was not uniformly compressed. “ the goal is not to reduce “intelligence,” but to allocate compute more effectively.” The answer cleared up my doubts.
> >
> > I understand that I make short inferences (not stopping thinking) when I am confident, and I make long inferences when I am low. I understand that when you are in high trust, you can go back to detailed reasoning, and clearly have room for reconsideration and correction, but small (not that you are no longer thinking. **To be sure, if it is high trust for the LLM, why can the reconsideration and revision be clearly reduced?** I want to know the logic, not the evidence on the benchmark.
> >
> > Thanks for the pass@k results. I was satisfied with the sufficient performance improvement in Pass@k in AIME. In addition to the standard method, we recommend that you add this result to the reader more kindly to the table.
> >
> > I thought it was a very interesting result together with the above results. Statistically, it was confirmed that the number of tokens was decreasing overall in sufficient situations and was not pulled to extreme values. I think this is important to clear up the reader's doubts, so I recommend adding it.

---

> > > ### Author Response · Authors · 2026-04-06
> > >
> > > Thanks the reviewer for the follow-up discussion. We are glad that our previous responses have resolved your concerns. To our understanding, the remaining question now is about the motivation for switching to concise mode in high-confidence regions, specifically: **if the LLM is in a high-trust state, why is it reasonable to reduce reconsideration and revision?** We clarify this intuition below.
> > >
> > > The key logic is high local confidence indicates that, conditional on the current reasoning state, the next continuation is relatively settled: the model is usually no longer choosing among several competing reasoning directions, but is instead executing an already selected plan (e.g., routine algebra, substitution, simplification, or summarization). In such regions, additional reconsideration is  redundant and has lower expected value because it is **less likely to change the reasoning trajectory in a meaningful way**. By contrast, low-confidence regions indicate that multiple plausible continuations are still active, i.e., the reasoning process is at a local decision point. Extra computation is therefore more useful there, because it can still revise assumptions, switch branches, or correct an emerging error.
> > >
> > > This intuition is also consistent with [1, 2], which suggests that low-confidence tokens correspond to critical reasoning forks or pivotal decision points, whereas high-confidence tokens more often complete an already ongoing linguistic or deterministic structure. In this sense, **the reasoning gains induced by RL are concentrated disproportionately on low-confidence “forking” tokens**, which means extened reasoning improve the model mostly on the low-confidence tokens that determine reasoning direction. This provides further justification for reducing compute in high-confidence regions, since they are less likely to contain the decision points that are most critical to successful reasoning.
> > >
> > > **Reference**
> > >
> > > [1] Wang et al. Beyond the 80/20 Rule: High-Entropy Minority Tokens Drive Effective Reinforcement Learning for LLM Reasoning
> > >
> > > [2] Cui et al. The Entropy Mechanism of Reinforcement Learning for Reasoning Language Models

---

### Decision · Program_Chairs · 2026-04-30

**Decision:**

Accept (regular)

**Comment:**

This submission studies an important and timely problem in reasoning models: not all parts of a chain of thought require the same amount of deliberation, so applying uniformly long-form reasoning can introduce unnecessary cost and redundancy. The paper proposes MixReasoning, a method that dynamically switches between detailed and concise reasoning modes within a single response using a concise LoRA adapter and a confidence-based controller. Overall, I find the core idea interesting, technically meaningful, and relevant to the growing literature on efficient reasoning.

Reviewer feedback was mixed, but the rebuttal meaningfully improved the picture. Reviewer cGmq explicitly stated that their concerns were fully resolved and raised their score to accept, noting that the method is promising and the results are good enough to warrant acceptance. Reviewer zPQD also indicated that the major concerns were resolved, with only a follow-up conceptual question remaining, and in the final justification updated the score upward to weak accept. These two reviewers both viewed the work as a useful contribution to efficient chain-of-thought reasoning.

The remaining concerns primarily come from reviewers uRdy and YdxE. For YdxE, the initial issues involved motivation, breadth of evaluation, and practical implications. In rebuttal, the authors provided additional discussion of the confidence-based motivation, expanded the evaluation to more tasks and model families, and discussed practical applications and possible extensions to training. YdxE still maintained a weak reject, mainly citing variance and limited gains on some tasks. I agree the gains are not uniformly large, and the paper is better framed as preserving performance while reducing reasoning cost rather than substantially improving accuracy. However, I do not view this as disqualifying given the paper’s stated objective and the additional variance reporting in the response.

Reviewer uRdy’s core concern is practical efficiency under real inference systems, especially switching overhead, rollback cost, KV-cache handling, and throughput implications. This is the most substantive remaining issue. I do not think it outweighs the paper’s contribution for two reasons. First, the authors did provide a concrete implementation discussion and an ablation on rollback, and they clarified that KV caches are mode-specific rather than mixed. Second, and importantly, the reviewer’s final justification appears to continue from a premise that the paper relies on token-count reduction alone or on “single sample latency,” whereas the submission and rebuttal state that end-to-end latency with vLLM is already reported and that the reported measurements are not merely token counts. I therefore discount this concern somewhat, because at least part of the disagreement seems to arise from a mismatch between the reviewer’s characterization and the evidence the authors say is already in the paper and rebuttal. At the same time, I agree that a stronger throughput-oriented systems evaluation would improve the camera-ready and make the practical efficiency claim more convincing.

On balance, I find the paper technically sound enough for acceptance. The central idea is novel and likely to be of interest to the community working on efficient reasoning and controllable inference-time compute. The rebuttal addressed many concrete concerns with added evidence, and two reviewers ended in support. The remaining weaknesses especially around stronger systems-level validation and the fact that accuracy gains are modest, are real, but in my judgment they are better viewed as limitations of scope rather than fatal flaws. For these reasons, I recommend Weak Accept.